# Identifying Causal Effects Under Functional Dependencies

**Yizuo Chen**
Department of Computer Science
University of California, Los Angeles
Los Angeles, CA 90024
yizuo.chen@ucla.edu

**Adnan Darwiche**
Department of Computer Science
University of California, Los Angeles
Los Angeles, CA 90024
darwiche@cs.ucla.edu

## Abstract

We study the identification of causal effects, motivated by two improvements to identifiability which can be attained if one knows that some variables in a causal graph are functionally determined by their parents (without needing to know the specific functions). First, an unidentifiable causal effect may become identifiable when certain variables are functional. Second, certain functional variables can be excluded from being observed without affecting the identifiability of a causal effect, which may significantly reduce the number of needed variables in observational data. Our results are largely based on an elimination procedure which removes functional variables from a causal graph while preserving key properties in the resulting causal graph, including the identifiability of causal effects.

## 1 Introduction

A causal effect measures the impact of an intervention on some events of interest, and is exemplified by the question "what is the probability that a patient would recover had they taken a drug?" This type of question, also known as an interventional query, belongs to the second rung of Pearl's causal hierarchy [1] so it ultimately requires experimental studies if it is to be estimated from data. However, it is well known that such interventional queries can sometimes be answered based on observational queries (first rung of the causal hierarchy) which can be estimated from observational data. This becomes very significant when experimental studies are either not available, expensive to conduct, or would entail ethical concerns. Hence, a key question in causal inference asks when and how a causal effect can be estimated from available observational data assuming a causal graph is provided [2].

More precisely, given a set of *treatment* variables $\mathbf{X}$ and a set of *outcome* variables $\mathbf{Y}$, the causal effect of $\mathbf{x}$ on $\mathbf{Y}$, denoted $\Pr(\mathbf{Y}|do(\mathbf{x}))$ or $\Pr_{\mathbf{x}}(\mathbf{Y})$, is the marginal probability on $\mathbf{Y}$ when an intervention sets the states of variables $\mathbf{X}$ to $\mathbf{x}$. The problem of identifying a causal effect studies whether $\Pr_{\mathbf{x}}(\mathbf{Y})$ can be uniquely determined from a causal graph and a distribution $\Pr(\mathbf{V})$ over some variables $\mathbf{V}$ in the causal graph [2], where $\Pr(\mathbf{V})$ is typically estimated from observational data. The casual effect is guaranteed to be identifiable if $\mathbf{V}$ correspond to all variables in the causal graph (with some positivity assumptions); that is, if all such variables are observed. When some variables are hidden (unobserved), it is possible that different parameterizations of the causal graph will induce the same distribution $\Pr(\mathbf{V})$ but different values for the causal effect $\Pr_{\mathbf{x}}(\mathbf{Y})$ which leads to unidentifiablility. In the past few decades, a significant amount of effort has been devoted to studying the identifiability of causal effects; see, e.g., [3, 2, 4–7]. Some early works include the *back-door criterion* [8, 2] and the *front-door criterion* [3, 2]. These criteria are sound but incomplete as they may fail to identify certain causal effects that are indeed identifiable. Complete identification methods include the do-calculus [2], the identification algorithm in [9], and the ID algorithm proposed in [10]. These methods require some positivity assumptions (constraints) on the observational distribution $\Pr(\mathbf{V})$ and can derive

an identifying formula that computes the causal effect based on $\Pr(\mathbf{V})$ when the causal effect is identifiable. Some recent works take a different approach by first estimating the parameters of a causal graph to obtain a fully-specified causal model which is then used to estimate causal effects through inference [11–14]. Further works focus on the efficiency of estimating causal effects from finite data [15–18], the general identifiability of causal effects from both observational and experimental data [19], and the causal effect identification with data collected from sub-populations [20].

A recent line of work studies the impact of additional information on identifiability, beyond causal graphs and observational data. For example, [21] showed that certain unidentifiable causal effects can become identifiable given information about context-specific independence. Our work in this paper follows the same direction as we consider the problem of causal effect identification in the presence of a particular type of qualitative knowledge called *functional dependencies* [22]. We say there is a functional dependency between a variable $X$ and its parents $\mathbf{P}$ in the causal graph if the distribution $\Pr(X|\mathbf{P})$ is deterministic but we do not know the distribution itself (i.e., the specific values of $\Pr(x|\mathbf{p})$). We also say in this case that variable $X$ is *functional.* Previous works have shown that functional dependencies can be exploited to improve the efficiency of Bayesian network inference [23, 13, 24–26]. We complement these works by showing that functional dependencies can also be exploited for identifiability. In particular, we show that some unidentifiable causal effects may become identifiable given such dependencies, propose techniques for testing identifiability in this context, and highlight other implications of such dependencies on the practice of identifiability.

Consider the following motivational example where we are interested in how the enforcement of speed limits may affect car accidents. The Driving Age ($A$) is functionally determined by Country ($C$); Driving Age and Country are causes of Speed ($X$); and Speed and Driving Age are causes of Accidents ($Y$). The DAG on the right captures the causal relations among these variables, where variable $A$ is circled to indicate it is functional. Suppose further that variables $C, X, Y$ are observed. According to classical causal-effect identification methods (e.g., do-calculus, ID algorithm), the causal effect of $X$ on $Y$ is unidentifiable in this case. However, if we take into account that variable $A$ is a function of $C$, which restricts the class of distributions under consideration, then the causal effect of $X$ on $Y$ becomes identifiable. This exemplifies the improvements to identifiability pursued in this paper.

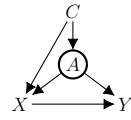

Consider a causal graph $G$ and a distribution $\Pr(\mathbf{V})$ over the observed variables $\mathbf{V}$ in $G$. To check the identifiability of a causal effect, it is standard to first apply the *projection* operation in [27, 28] which constructs another causal graph $G'$ with $\mathbf{V}$ as its non-root variables, and follow by applying an identification algorithm to $G'$ like the ID algorithm [10]. This two-stage procedure, which we will call *project-ID,* is applicable only under some positivity constraints (assumptions) which preclude some events from having a zero probability. Since such positivity constraints may contradict functional dependencies, we formulate the notion of *constrained identifiability* which takes positivity constraints as an input (in addition to the causal graph $G$ and distribution $\Pr(\mathbf{V})$). We also formulate the notion of *functional identifiability* which further takes functional dependencies as an input. This allows us to explicitly treat the interactions between positivity constraints and functional dependencies, which is needed for combining classical methods like project-ID with the results we present in this paper.

We start with some technical preliminaries in Section 2. We formally define positivity constraints and functional dependencies in Section 3 where we also introduce the problems of constrained and functional identifiability. Section 4 introduces two primitive operations, *functional elimination* and *functional projection,* which are needed for later treatments. Sections 5 presents our core results on functional identifiability and how they can be combined with existing identifiability algorithms. We finally close with concluding remarks in Section 6. Proofs of all results are included in the Appendix.

## 2 Technical Preliminaries

We consider discrete variables in this work. Single variables are denoted by uppercase letters (e.g., $X$) and their states are denoted by lowercase letters (e.g., $x$). Sets of variables are denoted by bold uppercase letters (e.g., $\mathbf{X}$) and their instantiations are denoted by bold lowercase letters (e.g., $\mathbf{x}$).

## 2.1 Causal Bayesian Networks and Interventions

A Causal Bayesian network (CBN) is a pair $\langle G, \mathcal{F} \rangle$ where $G$ is a *causal graph* in the form of a directed acyclic graph (DAG), and $\mathcal{F}$ is a set of conditional probability tables (CPTs). We have one CPT for each variable $X$ with parents $\mathbf{P}$ in $G$, which specifies the conditional probability distributions $\Pr(X|\mathbf{P})$. This CPT will often be denoted by $f_X(X, \mathbf{P})$ so $f_X(x, \mathbf{p}) \in [0, 1]$ for all instantiations $x, \mathbf{p}$ and $\sum_x f_X(x, \mathbf{p}) = 1$ for every instantiation $\mathbf{p}$.

A CBN induces a joint distribution over its variables $\mathbf{V}$ which is exactly the product of its CPTs, i.e., $\Pr(\mathbf{V}) = \prod_{V \in \mathbf{V}} f_V$. Applying a treatment $do(\mathbf{x})$ to the joint distribution yields a new distribution called the *interventional distribution,* denoted $\Pr_{\mathbf{x}}(\mathbf{V})$. One way to compute the interventional distribution is to consider the *mutilated CBN* $\langle G', \mathcal{F}' \rangle$ that is constructed from the original CBN $\langle G, \mathcal{F} \rangle$ as follows: remove from $G$ all edges that point to variables in $\mathbf{X}$, then replace the CPT in $\mathcal{F}$ for each $X \in \mathbf{X}$ with a CPT $f_X(X)$ where $f_X(x) = 1$ if $x$ is consistent with $\mathbf{x}$ and $f_X(x) = 0$ otherwise. Figure 1a depicts a causal graph $G$ and Figure 1b depicts the mutilated causal graph $G'$ under a treatment $do(x_1, x_2)$. The interventional distribution $\Pr_{\mathbf{x}}$ is the distribution induced by the mutilated CBN $\langle G', \mathcal{F}' \rangle$, where $\Pr_{\mathbf{x}}(\mathbf{Y})$ corresponds to the causal effect $\Pr(\mathbf{Y}|do(\mathbf{x}))$ also notated by $\Pr(\mathbf{Y}_{\mathbf{x}})$.

## 2.2 Identifying Causal Effects

A key question in causal inference is to check whether a causal effect can be (uniquely) computed given the causal graph $G$ and a distribution $\Pr(\mathbf{V})$ over a subset $\mathbf{V}$ of its variables. If the answer is yes, we say that the causal effect is *identifiable* given $G$ and $\Pr(\mathbf{V})$. Otherwise, the causal effect is *unidentifiable*. Variables $\mathbf{V}$ are said to be *observed* and the remaining variables are said to be *hidden*, where $\Pr(\mathbf{V})$ is usually estimated from observational data. We start with the general definition of identifiability (not necessarily for causal effects) from [2, Ch. 3.2.4] with a slight rephrasing.

**Definition 1** (Identifiability [2])**.** *Let $Q(M)$ be any computable quantity of a model $M$. We say that $Q$ is identifiable in a class of models if, for any pairs of models $M_1$ and $M_2$ from this class, $Q(M_1) = Q(M_2)$ whenever $\Pr_{M_1}(\mathbf{V}) = \Pr_{M_2}(\mathbf{V})$ where $\mathbf{V}$ are the observed variables.*

In the context of causal effects, the problem of identifiability is to check whether every pair of fully-specified CBNs ($M_1$ and $M_2$ in Definition 1) that induce the same distribution $\Pr(\mathbf{V})$ will also produce the same value for the causal effect. Note that Definition 1 does not restrict the considered models $M_1$ and $M_2$ based on properties of the distributions $\Pr_{M_1}(\mathbf{V})$ and $\Pr_{M_2}(\mathbf{V})$. However, in the literature on identifying causal effects, it is quite common to only consider CBNs (models) that induce distributions which satisfy some positivity constraints, such as $\Pr(\mathbf{V}) > 0$. We will examine such constraints more carefully in Section 3 as they may contradict functional dependencies.

It is well known that under some positivity constraints (e.g., $\Pr(\mathbf{V}) > 0$), the identifiability of causal effects can be efficiently tested using what we shall call the *project-ID* algorithm. Given a causal graph $G$, project-ID first applies the projection operation in [27–29] to yield a new causal graph $G'$ whose hidden variables are all roots and each has exactly two children. These properties are needed by the ID algorithm [10], which is then applied to $G'$ to yield an identifying formula if the causal effect is identifiable or FAIL otherwise. Consider the causal effect $\Pr_{x_1 x_2}(y)$ in Figure 1a where the only hidden variable is the non-root variable $B$. We first project the causal graph $G$ in Figure 1a onto its observed variables to yield the causal graph $G'$ in Figure 1c (all hidden variables in $G'$ are auxiliary and roots). We then run the ID algorithm on $G'$ which returns the following (simplified) identifying formula: $\Pr_{x_1 x_2}(y) = \sum_{acd} \Pr(c|x_1) \Pr(d|x_1, x_2) \sum_{x_1' x_2'} \Pr(y|x_1', x_2', a, c, d) \Pr(x_2'|x_1', a, c) \Pr(x_1', a)$. Hence, the causal effect $\Pr_{x_1 x_2}(y)$ is identifiable and can be computed using the above formula. Moreover, all quantities in the formula can be obtained from the distribution $\Pr(A, C, D, X_1, X_2, Y)$ over observed variables, which can be estimated from observational data. More details on the projection operation and the ID algorithm can be found in Appendix A.

## 3 Constrained and Functional Identifiability

As mentioned earlier, Definition 1 of identifiability [2, Ch. 3.2.4] does not restrict the pair of considered models $M_1$ and $M_2$. However, it is common in the literature on causal-effect identifiability to only consider CBNs with distributions $\Pr(\mathbf{V})$ that satisfy some positivity constraints. Strict positivity,

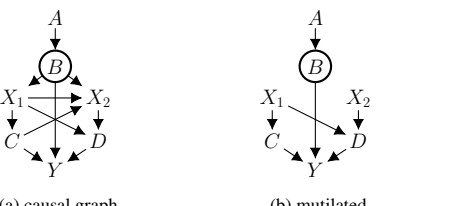

| (a) causal graph | (b) mutilated | (c) projected |

Figure 1: Example adapted from [30]. Hidden variables are circled. A bidirected edge $X \leftarrow\!\!\dashrightarrow Y$ is compact notation for $X \leftarrow H \rightarrow Y$ where $H$ is an auxiliary hidden variable.

$\Pr(\mathbf{V}) > 0$, is perhaps the mostly widely used constraint [9, 29, 2]. That is, in Definition 1, we only consider CBNs $M_1$ and $M_2$ which induce distributions $\Pr_{M_1}$ and $\Pr_{M_2}$ that satisfy $\Pr_{M_1}(\mathbf{V}) > 0$ and $\Pr_{M_2}(\mathbf{V}) > 0$. Weaker, and somewhat intricate, positivity constraints were employed by the ID algorithm in [10] as discussed in Appendix A, but we will apply this algorithm only under strict positivity to keep things simple. See also [31] for a recent discussion of positivity constraints.

Positivity constraints are motivated by two considerations: technical convenience, and the fact that most causal effects would be unidentifiable without some positivity constraints (more on this later). Given the multiplicity of positivity constraints considered in the literature, and given the subtle interaction between positivity constraints and functional dependencies (which are the main focus of this work), we next provide a systematic treatment of identifiability under positivity constraints.

### 3.1 Positivity Constraints

We will first formalize the notion of a *positivity constraint* and then define the notion of *constrained identifiability* which takes a set of positivity constraints as input (in addition to the causal graph $G$ and distribution $\Pr(\mathbf{V})$).[1]

**Definition 2.** *A positivity constraint on* $\Pr(\mathbf{V})$ *is an inequality of the form* $\Pr(\mathbf{S}|\mathbf{Z}) > 0$ *where* $\mathbf{S} \subseteq \mathbf{V}$, $\mathbf{Z} \subseteq \mathbf{V}$ *and* $\mathbf{S} \cap \mathbf{Z} = \emptyset$. *That is, for all instantiations* $\mathbf{s}, \mathbf{z}$, *if* $\Pr(\mathbf{z}) > 0$ *then* $\Pr(\mathbf{s}, \mathbf{z}) > 0$.

When $\mathbf{Z} = \emptyset$, the positivity constraint is defined on a marginal distribution, $\Pr(\mathbf{S}) > 0$. We may impose multiple positivity constraints on a set of variables $\mathbf{V}$. We will use $\mathcal{C}_\mathbf{V}$ to denote the set of positivity constraints imposed on $\Pr(\mathbf{V})$ and use $\texttt{vars}(\mathcal{C}_\mathbf{V})$ to denote all the variables mentioned by $\mathcal{C}_\mathbf{V}$. The weakest set of positivity constraints is $\mathcal{C}_\mathbf{V} = \{\}$ (no positivity constraints as in Definition 1), and the strongest positivity constraint is $\mathcal{C}_\mathbf{V} = \{\Pr(\mathbf{V}) > 0\}$.

We next provide a definition of identifiability for the causal effect of treatments $\mathbf{X}$ on outcomes $\mathbf{Y}$ in which positivity constraints are an input to the identifiability problem. We call it *constrained identifiability* in contrast to the (unconstrained) identifiability of Definition 1.

**Definition 3.** *We call* $\langle G, \mathbf{V}, \mathcal{C}_\mathbf{V} \rangle$ *an identifiability tuple where* $G$ *is a causal graph (DAG),* $\mathbf{V}$ *is its set of observed variables, and* $\mathcal{C}_\mathbf{V}$ *is a set of positivity constraints.*

**Definition 4** (Constrained Identifiability). *Let* $\langle G, \mathbf{V}, \mathcal{C}_\mathbf{V} \rangle$ *be an identifiability tuple. The causal effect of* $\mathbf{X}$ *on* $\mathbf{Y}$ *is said to be identifiable with respect to* $\langle G, \mathbf{V}, \mathcal{C}_\mathbf{V} \rangle$ *if* $\Pr_\mathbf{x}^1(\mathbf{y}) = \Pr_\mathbf{x}^2(\mathbf{y})$ *for any pair of distributions* $\Pr^1$ *and* $\Pr^2$ *which are induced by* $G$ *and that satisfy* $\Pr^1(\mathbf{V}) = \Pr^2(\mathbf{V})$ *and that also satisfy the positivity constraints* $\mathcal{C}_\mathbf{V}$.

For simplicity, we say "identifiability" to mean "constrained identifiability" in the rest of paper.

We next show that without some positivity constraints, most causal effects would not be identifiable. We say that a treatment $X \in \mathbf{X}$ is a *first ancestor* of some outcome $Y \in \mathbf{Y}$ if $X$ is an ancestor of $Y$ in causal graph $G$, and there exists a directed path from $X$ to $Y$ that is not intercepted by $\mathbf{X} \setminus \{X\}$. A first ancestor must exist if some treatment variable is an ancestor of some outcome variable.

**Proposition 5.** *The casual effect of* $\mathbf{X}$ *on* $\mathbf{Y}$ *is not identifiable wrt an identifiability tuple* $\langle G, \mathbf{V}, \mathcal{C}_\mathbf{V} \rangle$ *if some* $X \in \mathbf{X}$ *is a first ancestor of some* $Y \in \mathbf{Y}$, *and* $\mathcal{C}_\mathbf{V}$ *does not imply* $\Pr(X) > 0$.

---

[1]We are incorporating positivity constraints directly into the definition of identifiability. This is different from the analysis in [32] which derives positivity constraints from a particular run of the identification algorithm.

Hence, identifiability is not possible without some positivity constraints if at least one treatment variable is an ancestor of some outcome variable (which is common). Consider the causal graph on the right where $U$ is the only hidden variable. By Proposition 5, the causal effect of $\{X_1, X_2\}$ on $\{Y_1, Y_2\}$ is not identifiable if the considered distributions do not satisfy $\Pr(X_2) > 0$ as $X_2$ is a first ancestor of $Y_2$.

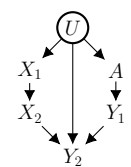

As positivity constraints become stronger, more causal effects become identifiable since the set of considered models becomes smaller. Consider the causal graph on the right in which all variables are observed, $\mathbf{V} = \{X, Y, Z\}$. Without positivity constraints, $\mathcal{C}_\mathbf{V} = \emptyset$, the causal effect of $X$ on $Y$ is not identifiable. However, it becomes identifiable given strict positivity, $\mathcal{C}_\mathbf{V} = \{\Pr(X, Y, Z) > 0\}$, leading to the identifying formula $\Pr_x(y) = \sum_z \Pr(y|x, z) \Pr(z)$. This causal effect is also identifiable under the weaker positivity constraint $\mathcal{C}_\mathbf{V} = \{\Pr(X|Z) > 0\}$,[2] which implies $\Pr(X) > 0$, so strict positivity is not necessary for identifiability even though it is typically assumed for this folklore result. This is an example where strict positivity may be assumed for technical convenience only as it may facilitate the application of some identifiability techniques like the do-calculus [2].

## 3.2 Functional Dependencies

A variable $X$ in a causal graph is said to *functionally depend* on its parents $\mathbf{P}$ if its distribution is deterministic: $\Pr(x|\mathbf{p}) \in \{0, 1\}$ for every instantiation $x, \mathbf{p}$. Variable $X$ is also said to be *functional* in this case. In this work, we assume *qualitative* functional dependencies: *we do not know the distribution* $\Pr(X|\mathbf{P})$, *we only know that it is deterministic.*[3]

The table on the right shows two variables $B$ and $C$ that both have $A$ as their parent. Variable $C$ is functional but variable $B$ is not. The CPT for variable $C$ will be called a *functional CPT* in this case. Functional CPTs are also known as (causal) mechanisms and are expressed using structural equations in

| $A$ | $B$ | $C$ | $\Pr(B|A)$ | $\Pr(C|A)$ |
|-----|-----|-----|------------|------------|
| 0 | 0 | 0 | 0.2 | 0 |
| 0 | 1 | 1 | 0.8 | 1 |
| 1 | 0 | 0 | 0.6 | 1 |
| 1 | 1 | 1 | 0.4 | 0 |

structural causal models (SCMs) [33–35]. By definition, in an SCM, every non-root variable is assumed to be functional (when noise variables are represented explicitly in the causal graph).

Qualitative functional dependencies are a longstanding concept. For example, they are common in relational databases, see, e.g., [36, 37], and their relevance to probabilistic reasoning had been brought up early in [22, Ch. 3]. One example of a (qualitative) functional dependency is that different countries have different driving ages, so we know that "Driving Age" functionally depends on "Country" even though we may not know the specific driving age for each country. Another example is that a letter grade for a class is functionally dependent on the student's weighted average even though we may not know the scheme for converting a weighted average to a letter grade.

In this work, we assume that we are given a causal graph $G$ in which some variables $\mathbf{W}$ have been designated as functional. The presence of functional variables further restricts the set of distributions $\Pr$ that we consider when checking identifiability. This leads to a more refined problem that we call *functional identifiability (F-identifiability)*, which depends on four elements.

**Definition 6.** *We call* $\langle G, \mathbf{V}, \mathcal{C}_\mathbf{V}, \mathbf{W} \rangle$ *an F-identifiability tuple when $G$ is a DAG, $\mathbf{V}$ is its set of observed variables, $\mathcal{C}_\mathbf{V}$ is a set of positivity constraints, and $\mathbf{W}$ is a set of functional variables in $G$.*

**Definition 7** (F-Identifiability). *Let* $\langle G, \mathbf{V}, \mathcal{C}_\mathbf{V}, \mathbf{W} \rangle$ *be an F-identifiability tuple. The causal effect of* $\mathbf{X}$ *on* $\mathbf{Y}$ *is* F-identifiable *wrt* $\langle G, \mathbf{V}, \mathcal{C}_\mathbf{V}, \mathbf{W} \rangle$ *if* $\Pr^1_\mathbf{x}(\mathbf{y}) = \Pr^2_\mathbf{x}(\mathbf{y})$ *for any pair of distributions* $\Pr^1$ *and* $\Pr^2$ *which are induced by $G$, and that satisfy* $\Pr^1(\mathbf{V}) = \Pr^2(\mathbf{V})$ *and the positivity constraints* $\mathcal{C}_\mathbf{V}$*, and in which variables* $\mathbf{W}$ *functionally depend on their parents.*

Both $\mathcal{C}_\mathbf{V}$ and $\mathbf{W}$ represent constraints on the models (CBNs) we consider when checking identifiability, and these two types of constraints may contradict each other. We next define two notions that characterize some important interactions between positivity constraints and functional variables.

---

[2]This weaker positivity constraint is sufficient to make the identifying formula well-defined since $\Pr(y|x, z) \Pr(z)$ in the formula is equal to zero when $\Pr(z) = 0$, and is computable when $\Pr(z) > 0$; that is, the conditional probability $\Pr(y|x, z)$ is well-defined if $\Pr(x|z) > 0$.

[3]We assume that root variables cannot be functional as such variables can be removed from the causal graph.

**Definition 8.** *Let $\langle G, \mathbf{V}, \mathcal{C}_{\mathbf{V}}, \mathbf{W} \rangle$ be an F-identifiability tuple. Then $\mathcal{C}_{\mathbf{V}}$ and $\mathbf{W}$ are consistent if there exists a parameterization for G which induces a distribution satisfying $\mathcal{C}_{\mathbf{V}}$ and in which $\mathbf{W}$ functionally depend on their parents. Moreover, $\mathcal{C}_{\mathbf{V}}$ and $\mathbf{W}$ are separable if $\mathbf{W} \cap \mathrm{vars}(\mathcal{C}_{\mathbf{V}}) = \emptyset$.*

If $\mathcal{C}_{\mathbf{V}}$ is inconsistent with $\mathbf{W}$ then the set of distributions $\mathrm{Pr}$ considered in Definition 7 is empty and, hence, the causal effect is not well defined (and trivially identifiable according to Definition 7). As such, one would usually want to ensure such consistency. Here are some examples of positivity constraints that are always consistent with a set of functional variables $\mathbf{W}$: positivity on each treatment variable, i.e., $\{\mathrm{Pr}(X) > 0, X \in \mathbf{X}\}$, positivity on the set of non-functional treatments, i.e., $\{\mathrm{Pr}(\mathbf{X} \setminus \mathbf{W}) > 0\}$, positivity on all non-functional variables, i.e., $\{\mathrm{Pr}(\mathbf{V} \setminus \mathbf{W}) > 0\}$. All these examples are special cases of the following condition. For a functional variable $W \in \mathbf{W}$, let $\mathbf{H}_W$ be variables that intercept all directed paths from non-functional variables to $W$ (such $\mathbf{H}_W$ may not be unique). If none of the positivity constraints in $\mathcal{C}_{\mathbf{V}}$ mentions both $W$ and $\mathbf{H}_W$, then $\mathcal{C}_{\mathbf{V}}$ and $\mathbf{W}$ are guaranteed to be consistent (see Proposition 25 in Appendix C).

Separability is a stronger condition and it intuitively implies that the positivity constraints will not rule out any possible functions for the variables in $\mathbf{W}$. We need such a condition for one of the results we present later. Some examples of positivity constraints that are separable from $\mathbf{W}$ are $\{\mathrm{Pr}(\mathbf{X} \setminus \mathbf{W}) > 0\}$ and $\{\mathrm{Pr}(\mathbf{V} \setminus \mathbf{W}) > 0\}$. Studying the interactions between positivity constraints and functional variables, as we did in this section, will prove helpful later when utilizing existing identifiability algorithms (which require positivity constraints) for testing functional identifiability.

## 4 Functional Elimination and Projection

Our approach for testing identifiability under functional dependencies will be based on eliminating functional variables from the causal graph, which may be followed by invoking the project-ID algorithm on the resulting graph. This can be subtle though since the described process will not work for every functional variable as we discuss in the next section. Moreover, one needs to handle the interaction between positivity constraints and functional variables carefully. The first step though is to formalize the process of eliminating a functional variable and to study the associated guarantees.

Eliminating variables from a probabilistic model is a well studied operation, also known as marginalization; see, e.g., [38–40]. When eliminating variable $X$ from a model that represents distribution $\mathrm{Pr}(\mathbf{Z})$, the goal is to obtain a model that represents the marginal distribution $\mathrm{Pr}(\mathbf{Y}) = \sum_x \mathrm{Pr}(x, \mathbf{Y})$ where $\mathbf{Y} = \mathbf{Z} \setminus \{X\}$. Elimination can also be applied to a DAG $G$ that represents conditional independencies $\mathcal{I}$, leading to a new DAG $G'$ that represents independencies $\mathcal{I}'$ that are implied by $\mathcal{I}$. In fact, the projection operation of [27, 28] we discussed earlier can be understood in these terms. We next propose an operation that eliminates functional variables from a DAG and that comes with stronger guarantees compared to earlier elimination operations as far as preserving independencies.

**Definition 9.** *The functional elimination of a variable $X$ from a DAG $G$ yields a new DAG attained by adding an edge from each parent of $X$ to each child of $X$ and then removing $X$ from $G$.[4]*

For convenience, we sometimes say "elimination" to mean "functional elimination" when the context is clear. From the viewpoint of independence relations, functional elimination is not sound if the eliminated variable is not functional. In particular, the DAG $G'$ that results from this elimination process may satisfy independencies (identified by d-separation) that do not hold in the original DAG $G$. As we show later, however, every independence implied by $G'$ must be implied by $G$ if the eliminated variable is functional. In the context of SCMs, functional elimination may be interpreted as replacing the eliminated variable $X$ by its function in all structural equations that contain $X$. Functional elimination applies in broader contexts than SCMs though. Eliminating multiple functional variables in any order yields the same DAG (see Proposition 22 in Appendix B). For example, eliminating variables $\{C, D\}$ from the DAG in Figure 2a yields the DAG in Figure 2c whether we use the order $\pi_1 = C, D$ or the order $\pi_2 = D, C$.

Functional elimination preserves independencies that hold in the original DAG and which are not preserved by other elimination methods including projection as defined in [27, 28]. These independencies are captured using the notion of D-separation [41, 42] which is more refined than the classical notion of d-separation [43, 44] (uppercase D- versus lowercase d-). The original definition

---

[4]Appendix B extends this definition to Causal Bayesian networks (i.e., updating both CPTs and causal graph).

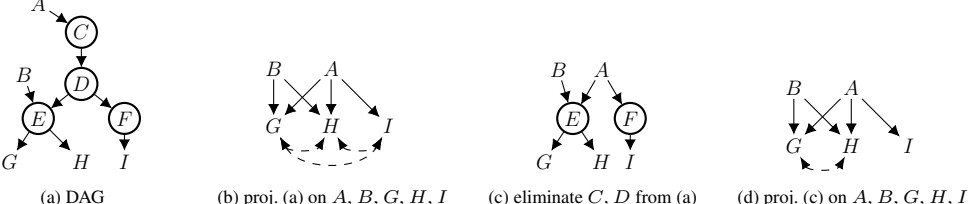

(a) DAG      (b) proj. (a) on $A, B, G, H, I$      (c) eliminate $C, D$ from (a)      (d) proj. (c) on $A, B, G, H, I$

Figure 2: Contrasting projection with functional projection. $C, D$ are functional. Hidden variables are circled.

of D-separation can be found in [42]. We provide a simpler definition next, stated as Proposition 10, as the equivalence between the two definitions is not immediate.

**Proposition 10.** *Let $\mathbf{X}, \mathbf{Y}, \mathbf{Z}$ be disjoint variable sets and $\mathbf{W}$ be a set of functional variables in DAG $G$. Then $\mathbf{X}$ and $\mathbf{Y}$ are D-separated by $\mathbf{Z}$ in $\langle G, \mathbf{W} \rangle$ iff $\mathbf{X}$ and $\mathbf{Y}$ are d-separated by $\mathbf{Z}'$ in $G$ where $\mathbf{Z}'$ is obtained as follows. Initially, $\mathbf{Z}' = \mathbf{Z}$. Repeat the next step until $\mathbf{Z}'$ stops changing: add to $\mathbf{Z}'$ every variable in $\mathbf{W}$ whose parents are in $\mathbf{Z}'$.*

To illustrate the difference between d-separation and D-separation, consider again the DAG in Figure 2a and assume that variables $C, D$ are functional. Variable $G$ and $I$ are not d-separated by $A$ but they are D-separated by $A$. That is, there are distributions which are induced by the DAG in Figure 2a and in which $G$ and $I$ are not independent given $A$. However, $G$ and $I$ are independent given $A$ in every induced distribution in which the variables $C, D$ are functionally determined by their parents. Functional elimination preserves D-separation in the following sense.

**Theorem 11.** *Consider a DAG $G$ with functional variables $\mathbf{W}$. Let $G'$ be the result of functionally eliminating variables $\mathbf{W}' \subseteq \mathbf{W}$ from $G$. For any disjoint sets $\mathbf{X}, \mathbf{Y}, \mathbf{Z}$ in $G'$, $\mathbf{X}$ and $\mathbf{Y}$ are D-separated by $\mathbf{Z}$ in $\langle G, \mathbf{W} \rangle$ iff $\mathbf{X}$ and $\mathbf{Y}$ are D-separated by $\mathbf{Z}$ in $\langle G', \mathbf{W} \setminus \mathbf{W}' \rangle$.*

We now define the operation of functional projection which augments the original projection operation in [27, 28] in the presence of functional dependencies.

**Definition 12.** *Let $G$ be a DAG, $\mathbf{V}$ be its observed variables, and $\mathbf{W}$ be its hidden functional variables ($\mathbf{W} \cap \mathbf{V} = \emptyset$). The functional projection of $G$ on $\mathbf{V}$ is a DAG obtained by functionally eliminating variables $\mathbf{W}$ from $G$ then projecting the resulting DAG on variables $\mathbf{V}$.*

We will now contrast functional projection and classical projection using the causal graph in Figure 2a, assuming that the observed variables are $\mathbf{V} = \{A, B, G, H, I\}$ and the functional variables are $\mathbf{W} = \{C, D\}$. Applying classical projection to this causal graph yields the causal graph in Figure 2b. To apply functional projection, we first functionally eliminate $C, D$ from Figure 2a, which yields Figure 2c, then project Figure 2c on variables $\mathbf{V}$ which yields the causal graph in Figure 2d. So we now need to contrast Figure 2b (classical projection) with Figure 2d (functional projection). The latter is a strict subset of the former as it is missing two bidrected edges. One implication of this is that variables $G$ and $I$ are not d-separated by $A$ in Figure 2b because they are not d-separated in Figure 2a. However, they are D-separated in Figure 2a and hence they are d-separated in Figure 2d. So functional projection yielded a DAG that exhibits more independencies. Again, this is because $G$ and $I$ are D-separated by $A$ in the original DAG, a fact that is not visible to projection but is visible to (and exploitable by) functional projection.

An important corollary of functional projection is the following. Suppose all functional variables are hidden, then two observed variables are *D-separated* in the causal graph $G$ iff they are *d-separated* in the functional-projected graph $G'$. This shows that such D-separations in $G$ appear as classical d-separations in $G'$ which allows us to feed $G'$ into existing identifiability algorithms as we show later. This is a key enabler of some results we shall present next on testing functional identifiability.

## 5   Causal Identification with Functional Dependencies

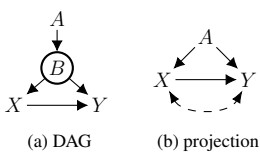

(a) DAG      (b) projection

Figure 3: $B$ is functional.

Consider the causal graph $G$ in Figure 3a and let $\mathbf{V} = \{A, X, Y\}$ be its observed variables. According to Definition 4 of identifiability, the causal effect of $X$ on $Y$ is not identifiable with respect to $\langle G, \mathbf{V}, \mathcal{C}_{\mathbf{V}} \rangle$ where $\mathcal{C}_{\mathbf{V}} = \{\Pr(A, X, Y) > 0\}$. We can show this by projecting the

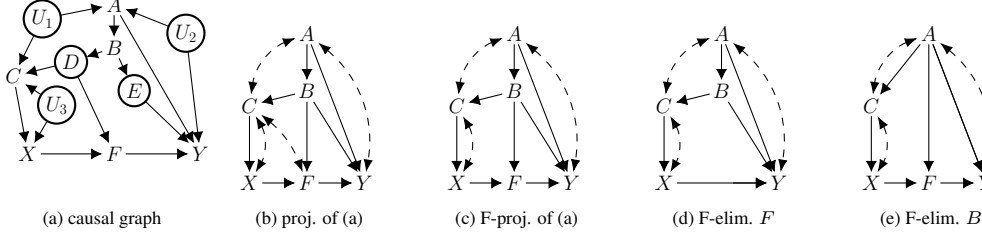

Figure 4: Variables $A, B, C, F, X, Y$ are observed. Variables $D, E$ are functional (and hidden).

causal graph $G$ on the observed variables $\mathbf{V}$, which yields the causal graph $G'$ in Figure 3b, then applying the ID algorithm to $G'$ which returns FAIL. Suppose now that the hidden variable $B$ is known to be functional. According to Definition 7 of F-identifiability, this additional knowledge reduces the number of considered models so it actually renders the causal effect identifiable — the identifying formula is $\Pr_x(y) = \sum_a \Pr(a) \Pr(y|a, x)$ as we show later. Hence, an unidentifiable causal effect became identifiable in light of knowledge that some variable is functional even without knowing the structural equations for this variable.

The question now is: How do we algorithmically test F-identifiablity? We will propose two techniques for this purpose, the first of which is geared towards exploiting existing algorithms for classical identifiability. This technique is based on eliminating functional variables from the causal graph while preserving F-identifiability, with the goal of getting to a point where F-identifiability becomes equivalent to classical identifiability. If we reach this point, we can use existing algorithms for classical identifiability, like the ID algorithm, to test F-identifiability. This can be subtle though since hidden functional variables behave differently from observed ones. We start with the following result.

**Theorem 13.** *Let $\langle G, \mathbf{V}, \mathcal{C}_{\mathbf{V}}, \mathbf{W} \rangle$ be an F-identifiability tuple. If $G'$ is the result of functionally eliminating the hidden functional variables $(\mathbf{W} \setminus \mathbf{V})$ from $G$, then the causal effect of $\mathbf{X}$ on $\mathbf{Y}$ is F-identifiable wrt $\langle G, \mathbf{V}, \mathcal{C}_{\mathbf{V}}, \mathbf{W} \rangle$ iff it is F-identifiable wrt $\langle G', \mathbf{V}, \mathcal{C}_{\mathbf{V}}, \mathbf{V} \cap \mathbf{W} \rangle$.*

An immediate corollary of this theorem is that if all functional variables are hidden, then we can reduce the question of F-identifiability to a question of identifiability since $\mathbf{V} \cap \mathbf{W} = \emptyset$ so F-identifiability wrt $\langle G', \mathbf{V}, \mathcal{C}_{\mathbf{V}}, \mathbf{V} \cap \mathbf{W} = \emptyset \rangle$ collapses into identifiability wrt $\langle G', \mathbf{V}, \mathcal{C}_{\mathbf{V}} \rangle$.

**Corollary 14.** *Let $\langle G, \mathbf{V}, \mathcal{C}_{\mathbf{V}}, \mathbf{W} \rangle$ be an F-identifiability tuple where $\mathcal{C}_{\mathbf{V}} = \{\Pr(\mathbf{V}) > 0\}$ and $\mathbf{W}$ are all hidden. If $G'$ is the result of functionally projecting $G$ on variables $\mathbf{V}$, then the causal effect of $\mathbf{X}$ on $\mathbf{Y}$ is F-identifiable wrt $\langle G, \mathbf{V}, \mathcal{C}_{\mathbf{V}}, \mathbf{W} \rangle$ iff it is identifiable wrt $\langle G, \mathbf{V}, \mathcal{C}_{\mathbf{V}} \rangle$.*[5]

This corollary suggests a method for using the ID algorithm, which is popular for testing identifiability, to establish F-identifiability by coupling ID with functional projection instead of classical projection. Consider the causal graph $G$ in Figure 4a with observed variables $\mathbf{V} = \{A, B, C, F, X, Y\}$. The causal effect of $X$ on $Y$ is not identifiable under $\Pr(\mathbf{V}) > 0$: projecting $G$ on observed variables $\mathbf{V}$ yields the causal graph $G'$ in Figure 4b and the ID algorithm produces FAIL on $G'$. Suppose now that the hidden variables $\{D, E\}$ are functional. To test whether the causal effect is F-identifiable using Corollary 14, we functionally project $G$ on the observed variables $\mathbf{V}$ which yields the causal graph $G''$ in Figure 4c. Applying the ID algorithm to $G''$ produces the following identifying formula: $\Pr_x(y) = \sum_{bf} \Pr(f|b, x) \sum_{acx'} \Pr(y|a, b, c, f, x') \Pr(a, b, c, x')$ so $\Pr_x(y)$ is F-identifiable.

We stress again that Corollary 14 and the corresponding F-identifiability algorithm apply only when all functional variables are hidden. We now treat the case when some of the functional variables are observed. The subtlety here is that, unlike hidden functional variables, eliminating an observed functional variable does not always preserve F-identifiability. However, the following result identifies conditions that guarantees the preservation of F-identifiability in this case. If all observed functional variables satisfy these conditions, then we can again reduce F-identifiability into identifiability so we can exploit existing methods for identifiability like the ID algorithm and do-calculus.

**Theorem 15.** *Let $\langle G, \mathbf{V}, \mathcal{C}_{\mathbf{V}}, \mathbf{W} \rangle$ be an F-identifiability tuple. Let $\mathbf{Z}$ be a set of observed functional variables that are neither treatments nor outcomes, are separable from $\mathcal{C}_{\mathbf{V}}$, and that have observed*

---

[5]We are requiring the positivity constraint $\Pr(\mathbf{V}) > 0$ as the projection operation in [28] requires it. If the projection operation only requires a weaker positivity constraint $\mathcal{C}'_{\mathbf{V}}$, we can replace $\mathcal{C}_{\mathbf{V}}$ by $\mathcal{C}'_{\mathbf{V}}$ in Corollary 14.

*parents. If $G'$ is the result of functionally eliminating variables $\mathbf{Z}$ from $G$, then the causal effect of $\mathbf{X}$ on $\mathbf{Y}$ is F-identifiable wrt $\langle G, \mathbf{V}, \mathcal{C}_{\mathbf{V}}, \mathbf{W} \rangle$ iff it is F-identifiable wrt $\langle G', \mathbf{V} \setminus \mathbf{Z}, \mathcal{C}_{\mathbf{V}}, \mathbf{W} \setminus \mathbf{Z} \rangle$.*

We now have the following important corollary of Theorems 13 & 15 which subsumes Corollary 14.

**Corollary 16.** *Let $\langle G, \mathbf{V}, \mathcal{C}_{\mathbf{V}}, \mathbf{W} \rangle$ be an F-identifiability tuple where $\mathcal{C}_{\mathbf{V}} = \{\Pr(\mathbf{V} \setminus \mathbf{W}) > 0\}$ and every variable in $\mathbf{W} \cap \mathbf{V}$ satisfies the conditions of Theorem 15. If $G'$ is the result of functionally projecting $G$ on $\mathbf{V} \setminus \mathbf{W}$, then the causal effect of $\mathbf{X}$ on $\mathbf{Y}$ is F-identifiable wrt $\langle G, \mathbf{V}, \mathcal{C}_{\mathbf{V}}, \mathbf{W} \rangle$ iff it is identifiable wrt $\langle G', \mathbf{V} \setminus \mathbf{W}, \mathcal{C}_{\mathbf{V}} \rangle$.*

Consider again the causal effect of $X$ on $Y$ in graph $G$ of Figure 4a with observed variables $\mathbf{V} = \{A, B, C, F, X, Y\}$. Suppose now that the observed variable $F$ is also functional (in addition to the hidden functional variables $D, E$) and assume $\Pr(A, B, C, X, Y) > 0$. Using Corollary 16, we can functionally project $G$ on $A, B, C, X, Y$ to yield the causal graph $G'$ in Figure 4d, which reduces F-identifiability on $G$ to classical identifiability on $G'$. Since strict positivity holds in $G'$, we can apply any existing identifiability algorithm and conclude that the causal effect is not identifiable. For another scenario, suppose that the observed variable $B$ (instead of $F$) is functional and we have $\Pr(A, C, F, X, Y) > 0$. Again, using Corollary 16, we functionally project $G$ onto $A, C, F, X, Y$ to yield the causal graph $G''$ in Figure 4e, which reduces F-identifiability on $G$ to classical identifiability on $G''$. If we apply the ID algorithm to $G''$ we get the identifying formula (which we denote as Eq. 1): $\Pr_x(y) = \sum_{af} \Pr(f|a, x) \sum_{cx'} \Pr(y|a, c, f, x') \Pr(a, c, x')$. In both scenarios above, we were able to test F-identifiability using an existing algorithm for identifiability.

Corollary 16 (and Theorem 15) has yet another key application: it can help us pinpoint observations that are not essential for identifiability. To illustrate, consider the second scenario above where the observed variable $B$ is functional in the causal graph $G$ of Figure 4a. The fact that Corollary 16 allowed us to eliminate variable $B$ from $G$ implies that observing this variable is not needed for rendering the causal effect F-identifiable and, hence, is not needed for computing the causal effect. This can be seen by examining the identifying formula (Eq. 1) which does not contain variable $B$. This application of Corollary 16 can be quite significant in practice, especially when some variables are expensive to measure (observe), or when they may raise privacy concerns; see, e.g., [45, 46].

Theorems 13 & 15 are more far-reaching than what the above discussion may suggest. In particular, even if we cannot eliminate every (observed) functional variable using these theorems, we may still be able to reduce F-identifiability to identifiability due to the following result.

**Theorem 17.** *Let $\langle G, \mathbf{V}, \mathcal{C}_{\mathbf{V}}, \mathbf{W} \rangle$ be an F-identifiability tuple. If every functional variable has at least one hidden parent, then a causal effect of $\mathbf{X}$ on $\mathbf{Y}$ is F-identifiable wrt $\langle G, \mathbf{V}, \mathcal{C}_{\mathbf{V}}, \mathbf{W} \rangle$ iff it is identifiable wrt $\langle G, \mathbf{V}, \mathcal{C}_{\mathbf{V}} \rangle$.*

That is, if we still have functional variables in the causal graph after applying Theorems 13 & 15, and if each such variable has at least one hidden parent, then F-identifiability is equivalent to identifiability.

The method we presented thus far for testing F-identifiability is based on eliminating functional variables from the causal graph, followed by applying existing tools for causal effect identification such as the project-ID algorithm and the do-calculus. This F-identifiability method is complete if every observed functional variable either satisfies the conditions of Theorem 15 or has at least one hidden parent that is not functional. This elimination-based method not only tests identifiability but also provides an identifying formula if the causal effect turns out to be identifiable.

We next present another technique for reducing F-identifiability to identifiability. This method is more general and much more direct than the previous one, but it does not allow us to fully exploit some existing tools like the ID algorithm due to the positivity assumptions they make. The new method is based on pretending that some of the hidden functional variables are actually observed and is inspired by Proposition 10 which reduces D-separation to d-separation using a similar technique.

**Theorem 18.** *Let $\langle G, \mathbf{V}, \mathcal{C}_{\mathbf{V}}, \mathbf{W} \rangle$ be an F-identifiability tuple where $\mathcal{C}_{\mathbf{V}} = \{\Pr(X) > 0, X \in \mathbf{X}\}$. A causal effect of $\mathbf{X}$ on $\mathbf{Y}$ is F-identifiable wrt $\langle G, \mathbf{V}, \mathcal{C}_{\mathbf{V}}, \mathbf{W} \rangle$ iff it is identifiable wrt $\langle G, \mathcal{C}_{\mathbf{V}}, \mathbf{V}' \rangle$ where $\mathbf{V}'$ is obtained as follows. Initially, $\mathbf{V}' = \mathbf{V}$. Repeat until $\mathbf{V}'$ stops changing: add to $\mathbf{V}'$ a functional variable from $\mathbf{W}$ if its parents are in $\mathbf{V}'$.*

Consider the causal effect of $X$ on $Y$ in graph $G$ of Figure 4a and suppose the observed variables are $\mathbf{V} = \{A, B, C, X, Y\}$, the functional variables are $\{D, E, F\}$ and we have $\Pr(X) > 0$. By Theorem 18, the causal effect of $X$ on $Y$ is F-identifiable iff it is identifiable in $G$ while pretending

that variables $\mathbf{V}' = \{A, B, C, D, E, F, X, Y\}$ are all observed. In this case, the casual effect is not identifiable but we cannot obtain this answer by applying an identifiability algorithm that requires positivity constraints which are stronger than $\Pr(X) > 0$. If we have stronger positivity constraints that imply $\Pr(X) > 0, X \in \mathbf{X}$, then only the if part of Theorem 18 will hold, assuming $\mathcal{C}_{\mathbf{V}}$ and $\mathbf{W}$ are consistent. That is, confirming identifiability wrt $\langle G, \mathcal{C}_{\mathbf{V}}, \mathbf{V}' \rangle$ will confirm F-identifiability wrt $\langle G, \mathbf{V}, \mathcal{C}_{\mathbf{V}}, \mathbf{W} \rangle$ but if identifiability is not confirmed then F-identifiability may still hold. This suggests that, to fully exploit the power of Theorem 18, one would need a new class of identifiability algorithms that can operate under the weakest possible positivity constraints.

## 6    Conclusion

We studied the identification of causal effects in the presence of a particular type of knowledge called functional dependencies. This augments earlier works that considered other types of knowledge such as context-specific independence. Our contributions include formalizing the notion of functional identifiability; the introduction of an operation for eliminating functional variables from a causal graph that comes with stronger guarantees compared to earlier elimination methods; and the employment (under some conditions) of existing algorithms, such as the ID algorithm, for testing functional identifiability and for obtaining identifying formulas. We also provided a complete reduction of functional identifiability to classical identifiability under very weak positivity constraints, and showed how our results can be used to reduce the number of variables needed in observational data.

## Acknowledgements

We wish to thank Scott Mueller, Jin Tian, and anonymous reviewers for providing valuable feedback on earlier versions of this paper. This work has been partially supported by ONR grant N000142212501.

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

# A  More On Projection and ID Algorithm

As mentioned in the main paper, the project-ID algorithm involves two steps: the projection operation and the ID algorithm. We will review more technical details of each step in this section.

## A.1  Projection

The projection [27–29] of $G$ onto $\mathbf{V}$ constructs a new DAG $G'$ over variables $\mathbf{V}$ as follows. Initially, DAG $G'$ contains variables $\mathbf{V}$ but no edges. Then for every pair of variables $X, Y \in \mathbf{V}$, an edge is added from $X$ to $Y$ to $G'$ if $X$ is a parent of $Y$ in $G$, or if there exists a directed path from $X$ to $Y$ in $G$ such that none of the internal nodes on the path is in $\mathbf{V}$. A bidirected edge $X \leftarrow\!\text{-}\!\text{-}\!\text{-}\!\rightarrow Y$ is further added between every pair of variables $X$ and $Y$ in $G'$ if there exists a divergent path[6] between $X$ and $Y$ in $G$ such that none of the internal nodes on the path is in $\mathbf{V}$. For example, the projection of the DAG in Figure 1a onto $A, C, D, X_1, X_2, Y$ yields Figure 1c. A bidirected edge $X \leftarrow\!\text{-}\!\text{-}\!\text{-}\!\rightarrow Y$ is compact notation for $X \leftarrow H \rightarrow Y$ where $H$ is an auxiliary hidden variable. Hence, the projected DAG in Figure 1c can be interpreted as a classical DAG but with additional, hidden root variables.

The projection operation is guaranteed to produce a DAG $G'$ in which hidden variables are all roots and each has exactly two children. Graphs that satisfy this property are called *semi-Markovian* and can be fed as inputs to the ID algorithm for testing identifiability [10]. Moreover, projection preserves some properties of $G$, such as d-separation [27] among $\mathbf{V}$, which guarantees that identifiabilty is preserved when working with $G'$ instead of $G$ [28].

## A.2  ID Algorithm

After obtaining a projected causal graph, we can apply the ID algorithm for identifying causal effects [10, 47]. The algorithm returns either an identifying formula if the causal effect is identifiable or FAIL otherwise. The algorithm is sound since each line of the algorithm can be proved with basic probability rules and do-calculus. The algorithm is also complete since a causal graph must contain a *hedge*, a graphical structure that induces the unidentifiability, if the algorithm returns FAIL. The algorithm, however, is only sound and complete under certain positivity constraints, which are weaker but more subtle than strict positivity $(\Pr(\mathbf{V}) > 0)$.

The positivity constraints required by ID can be summarized as follows ($\mathbf{X}$ are the treatment variables): (1) $\Pr(\mathbf{X}|\mathbf{P}) > 0$ where $\mathbf{P} = \text{Parents}(\mathbf{X}) \setminus \mathbf{X}$, and (2) $\Pr(\mathbf{Z}) > 0$ for all quantities $\Pr(\mathbf{S}|\mathbf{Z})$ considered by the ID algorithm. The second constraint depends on a particular run of the ID algorithm and can be interpreted as follows. First, if the ID algorithm returns FAIL then the causal effect is not identifiable even under the strict positivity constraint $\Pr(\mathbf{V}) > 0$. However, if the ID algorithm returns "identifiable" then the causal effect is identifiable under the above constraints which are now well defined given a particular run of the ID algorithm. We illustrate with an example next.

Consider the causal graph on the right which contains observed variables $\{A, B, C, X, Y\}$. Suppose we are interested in the causal effect of $X$ on $Y$, applying the ID algorithm returns the following identifying formula: $\Pr_x(y) = \sum_{abc} \Pr(c|a, b, x) \sum_{x'} \Pr(y|a, b, c, x') \Pr(a, b, x')$. The positivity constraint extracted from this run of the algorithm is $\Pr(a, b, c, x) > 0$ for all $a, b, c, x$. That is, we can only safely declare the causal effect identifiable based on the ID algorithm if this positivity constraint is satisfied.

# B  Functional Elimination for CBNs

The functional elimination in Definition 9 removes functional variables from a DAG $G$ and yields another DAG $G'$ on the remaining variables. We have shown in the main paper that the functional elimination preserves the D-separations. Here we extend the notion of functional elimination to Causal Bayesian networks (CBNs) which contain not only a causal graph (DAG) but also CPTs. We show that the (extended) functional elimination preserves the marginal distribution on the remaining variables. That is, given any CBN with the causal graph $G$, we can construct another CBN with the causal graph $G'$ such that the two CBNs induce a same distribution on the variables in $G'$, where $G'$ is the result of eliminating functional variables from $G$. Moreover, we show that the functional

---

[6]A divergent path between $X$ and $Y$ is a path in the form of $X \leftarrow \cdots \leftarrow U \rightarrow \cdots \rightarrow Y$.

elimination operation further preserves the causal effects, which makes it applicable to the causal identification. This extended version of functional elimination and the corresponding results will be used for the proofs in Appendix C.

Recall that a CBN $\langle G, \mathcal{F} \rangle$ contains a causal graph $G$ and a set of CPTs $\mathcal{F}$. We first extend the definition of functional elimination (Definition 9) from DAGs to CBNs.

**Definition 19.** *The functional elimination of a functional variable $X$ from a CBN $\langle G, \mathcal{F} \rangle$ yields another CBN $\langle G', \mathcal{F}' \rangle$ obtained as follows. The DAG $G'$ is obtained from $G$ by Definition 9. For each child $C$ of $X$, its CPT in $\mathcal{F}'$ is $(\sum_X f_X f_C)$ where $f_X$, $f_C$ are the corresponding CPTs in $\mathcal{F}$.*

We first show that the new CPTs produced by Definition 19 are well-defined.

**Proposition 20.** *Let $f_X$ and $f_Y$ be the CPTs for variables $X$ and $Y$ in a CBN, then $(\sum_X f_X f_Y)$ is a valid CPT for $Y$.*

The next proposition shows that functional elimination preserves the functional dependencies.

**Proposition 21.** *Let $\mathcal{M}'$ be the CBN resulting from functionally eliminating a functional variable from a CBN $\mathcal{M}$. Then each variable (from $\mathcal{M}'$) is functional in $\mathcal{M}'$ if it is functional in $\mathcal{M}$.*

The next theorem shows that the order of functional elimination does not matter.

**Proposition 22.** *Let $\mathcal{M}$ be a CBN and $\pi_1$, $\pi_2$ be two variable orders over a set of functional variables $\mathbf{W}$. Then functionally eliminating $\mathbf{W}$ from $\mathcal{M}$ according to $\pi_1$ and $\pi_2$ yield the same CBN.*

The next result shows that eliminating functional variables preserves the marginal distribution.

**Theorem 23.** *Consider a CBN $\mathcal{M}$ which induces $\Pr$. Let $\mathcal{M}'$ be the result of functionally eliminating a set of functional variables $\mathbf{W}$ from $\mathcal{M}$ which induces $\Pr'$. Then $\Pr' = \sum_{\mathbf{W}} \Pr$.*

One key property of functional elimination is that it preserves the interventional distribution over the remaining variables. This property allows us to eliminate functional variables from a causal graph and estimate the causal effects in the resulting graph.

**Theorem 24.** *Let $\mathcal{M}'$ be the CBN over variables $\mathbf{V}$ resulting from functionally eliminating a set of functional variables $\mathbf{W}$ from a CBN $\mathcal{M}$. Then $\mathcal{M}'$ and $\mathcal{M}$ attain the same $\Pr_{\mathbf{x}}(\mathbf{V})$ for any $\mathbf{X} \subseteq \mathbf{V}$.*

## C Proofs

The proofs of the results will be ordered slightly different from the order they appear in the main body of the paper.

**Proof of Proposition 5**

*Proof.* Our goal is to construct two different parameterizations $\mathcal{F}^1$ and $\mathcal{F}^2$ that induce the same $\Pr(\mathbf{V})$ but different $\Pr_{\mathbf{x}}(\mathbf{y})$. This is done by first creating a parameterization $\mathcal{F}$ which contains strictly positive CPTs for all variables, and then constructing $\mathcal{F}^1$ and $\mathcal{F}^2$ based on $\mathcal{F}$.

Let $\mathcal{P}$ be the directed path from $X = X_i$ to $Y$, denoted $X \to Z \to \cdots \to Y$ which does not contain any treatment variables other than $X$. Let $\mathbf{P}_X$ be the parents of $X$ in $G$. For each node $M$ on the path, let $\mathbf{P}_M$ be the parents of $M$ except for the parent that lies on $\mathcal{P}$. Moreover, for each variable $M$ on $\mathcal{P}$, we will only modify the conditional probability for a single state $m^*$ of $M$, where $x^* \in \mathbf{x}$ is the treated state of $X$. Let $\epsilon$ be an arbitrarily small constant (close to 0), we next show the modifications for the CPTs in $\mathcal{F}^1$.

$$f^1(x|\mathbf{p}_X) = \begin{cases} 0 & \text{if } x = x^* \\ 1/(|X| - 1), & \text{otherwise} \end{cases}$$

$$f^1(z|x, \mathbf{p}_Z) = \begin{cases} 1 - \epsilon, & \text{if } x = x^*, z = z^* \\ \epsilon/(|Z| - 1), & \text{if } x = x^*, z \neq z^* \\ \epsilon, & \text{if } x \neq x^*, z = z^* \\ (1 - \epsilon)/(|Z| - 1), & \text{if } x \neq x^*, z \neq z^* \end{cases}$$

For every variable $T \notin \{X, Z\}$ which has parent $Q$ on the path $\mathcal{P}$, we assign

$$f^1(t|q, \mathbf{p}_T) = \begin{cases} 1 - \epsilon, & \text{if } q = q^*, t = t^* \\ \epsilon/(|T| - 1), & \text{if } q = q^*, t \neq t^* \\ \epsilon, & \text{if } q \neq q^*, t = t^* \\ (1 - \epsilon)/(|T| - 1), & \text{if } q \neq q^*, t \neq t^* \end{cases}$$

We assign the same CPTs for $X$ and all variables $T \notin \{X, Z\}$ but a different CPT for $Z$ in $\mathcal{F}^2$.

$$f^2(z|x, \mathbf{p}_Z) = \begin{cases} \epsilon, & \text{if } x = x^*, z = z^* \\ (1 - \epsilon)/(|Z| - 1), & \text{if } x = x^*, z \neq z^* \\ \epsilon, & \text{if } x \neq x^*, z = z^* \\ (1 - \epsilon)/(|Z| - 1), & \text{if } x \neq x^*, z \neq z^* \end{cases}$$

The two parameterizations $\mathcal{F}^1$ and $\mathcal{F}^2$ induce the same $\Pr(\mathbf{V})$ where $\Pr(\mathbf{v}) = 0$ if $x^* \in \mathbf{v}$ and $\Pr(\mathbf{v}) > 0$ otherwise. We next show that the parameterization satisfies each positivity constraint $\Pr(\mathbf{S}|\mathbf{Z})$ as long as it does not imply $\Pr(X) > 0$. We first show that $X \in \mathbf{S}$ implies $\Pr(X) > 0$. This is because $\Pr(\mathbf{S}) = \sum_{\mathbf{z}} \Pr(\mathbf{S}|\mathbf{z}) \Pr(\mathbf{z})$ and there must exist some instantiation $\mathbf{z}$ where $\Pr(\mathbf{z}) > 0$ and $\Pr(\mathbf{S}|\mathbf{z}) > 0$ by constraint. This implies $\Pr(\mathbf{S}) > 0$ and therefore $\Pr(X) > 0$. Hence, $\mathcal{C}_\mathbf{V}$ does not contain such constraint $\Pr(\mathbf{S}|\mathbf{Z})$ where $X \in \mathbf{S}$. Suppose $X \in \mathbf{Z}$, then $\Pr(\mathbf{z}) > 0$ if and only if $x^* \notin \mathbf{z}$. Moreover, since $\Pr(\mathbf{v}) > 0$ whenever $x^* \notin \mathbf{v}$, it is guaranteed that $\Pr(\mathbf{S}, \mathbf{z}) > 0$ when $\Pr(\mathbf{z}) > 0$, which implies $\Pr(\mathbf{S}|\mathbf{Z}) > 0$. Finally, suppose $X \notin (\mathbf{S} \cup \mathbf{Z})$, then $\Pr(\mathbf{S}, \mathbf{Z}) = \sum_x \Pr(\mathbf{S}, \mathbf{Z}, x) > 0$. Hence, the positivity constraint is satisfied by both parameterizations. By construction, $\Pr^1$ and $\Pr^2$ induce different values for the causal effect $\Pr_\mathbf{x}(\mathbf{y})$ since the probability of $Y = y^*$ under the treatment $do(X = x^*)$ will be different for the two parameterizations. $\square$

**Proposition 25.** *Let $G$ be a causal graph and $\mathbf{V}$ be its observed variables. A set of functional variables $\mathbf{W}$ is consistent with positivity constraints $\mathcal{C}_\mathbf{V}$ if no single constraint in $\mathcal{C}_\mathbf{V}$ mentions both $W \in \mathbf{W}$ and a set $\mathbf{H}_W$ that intercepts all directed paths from non-functional variables to $W$.*

*Proof of Proposition 25.* We construct a parameterization $\mathcal{F}$ and show that the distribution $\Pr$ induced by $\mathcal{F}$ satisfies the $\mathcal{C}_\mathbf{V}$, which ensures the consistency. The states of each variable $V$ are represented in the form of $(s_V, p_1, \ldots, p_m)$ where $s_V$ and $p_i$ ($i \in \{1, \ldots, m\}$) are all binary indicator (0 or 1). Specifically, each of the $p_i$ corresponds to a "functional descendant paths" of $V$ defined as follows: a functional descendant path of $V$ is a directed path that starts with $V$ and that all variables on the path (excluding $V$) are functional. Suppose $V$ does not have any functional descendant paths, then the states of $V$ is simply represented as $(s_V)$.

We next show how to assign CPTs for each variable in the causal graph $G$ based on whether the variable is functional. For each non-functional variable, we assign a uniform distribution. For each functional variable $W$ whose parents are $T_1, \ldots, T_n$ and whose functional descendant paths are $\mathcal{P}_1, \ldots, \mathcal{P}_m$, we assign the CPT $f_W$ as follows:

$$\begin{aligned} s_W &\leftarrow p^{T_1}_{Ind(T_1, W)} \oplus \cdots \oplus p^{T_n}_{Ind(T_n, W)} \\ p_1 &\leftarrow p^{T_1}_{1,1} \oplus \cdots \oplus p^{T_n}_{n,1} \\ &\cdots \\ p_m &\leftarrow p^{T_1}_{1,m} \oplus \cdots \oplus p^{T_n}_{n,m} \end{aligned} \tag{1}$$

where $Ind(T_i, W)$ denotes the index assigned to the path $\{(T_i, W)\}$ (which contains a single edge) in the state of $T^i$, and $p^{T_i}_{i,j}$ denotes the indicator in the state of $T_i$ for the functional descendant path $\mathcal{P}'$ that contains the functional descendant path $\mathcal{P}_j$, i.e., $\mathcal{P}' = \{(T_i, W)\} \cup \mathcal{P}_j$.

For simplicity, we call the set of variables $\mathbf{H}_W$ that satisfies the condition in the proposition a "functional ancestor set" of $W$. We show that $\Pr(\mathbf{S}, \mathbf{Z}) > 0$ for each positivity constraint in the form of $\Pr(\mathbf{S}|\mathbf{Z}) > 0$. Let $\mathbf{W} \subseteq \mathbf{S} \cup \mathbf{Z}$ be a subset of functional variables. Since $\mathbf{S} \cup \mathbf{Z}$ does not contain any functional ancestor set of $W$ for each $W \in \mathbf{W}$, it follows that there exist directed paths from a set of non-functional variables $\mathcal{A}'_W$ to $W$ that are unblocked by $\mathbf{M} = \mathbf{S} \cup \mathbf{Z} \setminus \{W\}$ and contain only functional variables (excluding $\mathcal{A}'_W$). We can further assume that $\mathcal{A}'_W$ is chosen such that the set

$\mathcal{A}_W = \mathbf{M} \cup \mathcal{A}'_W$ forms a valid functional ancestor set for $W$. We next show that for any state $w$ of $W$ and instantiation $\mathbf{m}$ of $\mathbf{M}$, there exists at least one instantiation $\mathbf{a}$ of $\mathcal{A}'_W$ such that $\Pr(w, \mathbf{m}, \mathbf{a}) > 0$.

Let $\mathbf{P}^W$ denote the set of all directed paths from $\mathcal{A}_W$ to $W$ that do not contain $\mathcal{A}_W$ (except for the first node on the path). Let $\mathbf{P}_1^W \subseteq \mathbf{P}^W$ be the paths that start with a variable in $\mathbf{M}$, and $\mathbf{P}_2^W \subseteq \mathbf{P}^W$ be other paths that start with a variable in $\mathcal{A}'_W$. Moreover, for any path $\mathcal{P}$, let $\texttt{pathval}(\mathcal{P})$ be the binary indicator (e.g., $p_1$) for $\mathcal{P}$ in the state of $\mathcal{P}(0)$ (first variable in $\mathcal{P}$). Since the value assignments for $\texttt{pathval}(\mathcal{P})$ are independent for different $\mathcal{P}$'s, we can always find some instantiation $\mathbf{a} \in \mathcal{A}'_W$ such that the following equality holds given $w$ and $\mathbf{m}$:

$$\bigoplus_{\mathcal{P}_2 \in \mathbf{P}_2^W} \texttt{pathval}(\mathcal{P}_2) = s_W \oplus \bigoplus_{\mathcal{P}_1 \in \mathbf{P}_1^W} \texttt{pathval}(\mathcal{P}_1)$$

We next assign values for other path indicators of $\mathbf{a}$ such that the indicators for the functional descendant paths in the state $w$ are set correctly. In particular, for each functional descendant path $\mathcal{P}$ of $W$, let $\mathbf{P}$ be the set of functional descendant paths of $\mathcal{A}_W$ that do not contain $\mathcal{A}_W$ (except for the first node on the path) and that contain $\mathcal{P}$ as a sub-path. Let $\mathbf{P}_1 \subseteq \mathbf{P}$ be the paths that start with a variable in $\mathbf{M}$, and $\mathbf{P}_2 \subseteq \mathbf{P}$ be other paths that start with a variable in $\mathcal{A}'_W$. Again, since all the indicators for paths in $\mathcal{P}$ are independent, we can assign the indicators for $\mathbf{a} \in \mathcal{A}'_W$ such that

$$\bigoplus_{\mathcal{P}_2 \in \mathbf{P}_2} \texttt{pathval}(\mathcal{P}_2) = \texttt{pathval}(\mathcal{P}) \oplus \bigoplus_{\mathcal{P}_1 \in \mathbf{P}_1} \texttt{pathval}(\mathcal{P}_1)$$

Finally, we combine the cases for each individual $W \in \mathbf{W}$ by creating the following set $\mathcal{A}_{\mathbf{W}} = \bigcup_{W \in \mathbf{W}} \mathcal{A}_W$. Since all the functional descendant paths we considered for different $W$'s are disjoint, we can always find an assignment $\mathbf{a}$ for $\mathcal{A}_{\mathbf{W}}$ that is consistent with the functional dependencies (does not produce any zero probabilities). Consequently, there must exist some full instantiation $(\mathbf{u}, \mathbf{v})$ compatible with $\mathbf{s}, \mathbf{z}, \mathbf{a}$ such that $\Pr(\mathbf{u}, \mathbf{v}) > 0$, which implies $\Pr(\mathbf{s}, \mathbf{z}) > 0$. $\qquad\square$

## Proof of Proposition 20

*Proof.* Suppose $Y$ is not a child of $X$ in the CBN, then $\sum_X f_X f_Y = f_Y(\sum_X f_X)$ which is guaranteed to be a CPT for $Y$. Suppose $Y$ is a child of $X$. Let $\mathbf{P}_X$ denote the parents of $X$ and $\mathbf{P}_Y$ denote the parents of $Y$ excluding $X$. The new factor $g = \sum_X f_X f_Y$ is defined over $\mathbf{P}_X \cup \mathbf{P}_Y \cup \{Y\}$. Consider each instantiation $\mathbf{p}_X$ and $\mathbf{p}_Y$, then $\sum_y g(\mathbf{p}_X, \mathbf{p}_Y, y) = \sum_y \sum_x f_X(x|\mathbf{p}_x) f_Y(y|\mathbf{p}_Y, x) = \sum_x f_X(x|\mathbf{p}_X) \sum_y f_Y(y|\mathbf{p}_Y, x) = 1$. Hence, $g$ is a CPT for $Y$. $\qquad\square$

## Proof of Proposition 21

*Proof.* Let $X$ be the functional variables that is functionally eliminated. By definition, the elimination only affects the CPTs for the children of $X$. Hence, any functional variable that is not a child of $X$ remains functional. For each child $C$ of $X$ that is functional, the new CPT $(\sum_X f_X f_C)$ only contains values that are either 0 or 1 since both $f_X$ and $f_C$ are functional. $\qquad\square$

## Proof of Proposition 22

*Proof.* First note that $\pi_2$ can always be obtained from $\pi_1$ by a sequence of "transpositions", where each transposition swaps two adjacent variables in the first sequence. Let $\pi$ be an elimination order and let $\pi'$ be the elimination order resulted from swapping $\pi_i = X$ and $\pi_{i+1} = Y$ from the $\pi$, i.e.,

$$\pi = (\ldots, X, Y, \ldots) \qquad \pi' = (\ldots, Y, X, \ldots)$$

We show functional elimination according to $\pi$ and $\pi'$ yield a same CBN, which can be applied inductively to conclude that elimination according to $\pi_1$ and $\pi_2$ yield a same CBN. Since $\pi$ and $\pi'$ agree on a same elimination order up to $X$, they yield a same CBN before eliminating variables $X, Y$. It suffices to show the CBNs resulting from eliminating $(X, Y)$ and eliminating $(Y, X)$ are the same. Let $\langle G, \Pr \rangle$ be the CBN before eliminating variables $X, Y$. Suppose $X$ and $Y$ do not belong to a same family (which contains a variable and its parents), the elimination of $X$ and $Y$ are independent

and the order of elimination does not matter. Suppose $X$ and $Y$ belong to a same family, then they are either parent and child or co-parents.[7]

WLG, suppose $X$ is a parent of $Y$. Eliminating $(X, Y)$ and eliminating $(Y, X)$ yield a same causal graph that is defined as follows. Each child $C$ of $Y$ has parents $\mathbf{P}_X \cup \mathbf{P}_Y \cup \mathbf{P}_C \setminus \{X, Y\}$, and any other child $C$ of $X$ has parents $\mathbf{P}_X \cup \mathbf{P}_C \setminus \{X\}$. We next consider the CPTs. For each common child $C$ of $X$ and $Y$, its CPT resulting from eliminating $(X, Y)$ is $f_C^1 = \sum_Y (\sum_X f_C f_X)(\sum_X f_Y f_X)$, and the CPT resulting from eliminating $(Y, X)$ is $f_C^2 = \sum_X f_X (\sum_Y f_C f_Y)$. Since $X$ is a parent of $Y$, we have $Y \notin f_X$ and

$$
\begin{aligned}
f_C^2 &= \sum_X \sum_Y f_X f_C f_Y = \sum_Y \sum_X f_X f_C f_Y \\
&= \sum_Y (\sum_X f_X f_C)(\sum_X f_X f_Y) = f_C^1
\end{aligned}
$$

We next consider the case when $C$ is a child of $Y$ but not a child of $X$. The CPT for $C$ resulting from eliminating $(X, Y)$ is $f_C^1 = \sum_Y f_C (\sum_X f_Y f_X)$, and the CPT resulting from eliminating $(Y, X)$ is $f_C^2 = \sum_X f_X (\sum_Y f_C f_Y)$. Again, since $X$ is a parent of $Y$, we have $Y \notin f_X$ and

$$
\begin{aligned}
f_C^2 &= \sum_X \sum_Y f_X f_C f_Y = \sum_Y \sum_X f_X f_C f_Y \\
&= \sum_Y f_C (\sum_X f_X f_Y) = f_C^1
\end{aligned}
$$

We finally consider the case when $C$ is a child of $X$ but not a child of $Y$. Regardless of the order on $X$ and $Y$, the CPT for $C$ resulting from eliminating $X$ and $Y$ is $(\sum_X f_X f_C)$.

We next consider the case when $X$ and $Y$ are co-parents. Regardless of the order on $X$ and $Y$, the causal graph resulting from the elimination satisfies the following properties: (1) for each common child $C$ of $X$ and $Y$, the parents for $C$ are $\mathbf{P}_X \cup \mathbf{P}_Y \cup \mathbf{P}_C \setminus \{X, Y\}$; (2) for each $C$ that is a child of $X$ but not a child of $Y$, the parents for $C$ are $\mathbf{P}_X \cup \mathbf{P}_C \setminus \{X\}$; (3) for each $C$ that is a child of $Y$ but not a child of $X$, the parents for $C$ are $\mathbf{P}_Y \cup \mathbf{P}_C \setminus \{Y\}$. We next consider the CPTs. For each common child $C$ of $X$ and $Y$, its CPT resulting from eliminating $(X, Y)$ is $f_C^1 = \sum_Y f_Y (\sum_X f_X f_C)$, and the CPT resulting from eliminating $(Y, X)$ is $f_C^2 = \sum_X f_X (\sum_Y f_Y f_C)$. Since $X$ and $Y$ are not parent and child, we have $X \notin f_Y$, $Y \notin f_X$ and

$$
f_C^1 = \sum_Y \sum_X f_Y f_X f_C = \sum_X \sum_Y f_Y f_X f_C = f_C^2
$$

For each $C$ that is a child of $X$ but not a child of $Y$, regardless of the order on $X$ and $Y$, the CPT for $C$ resulting from eliminating variables $X$ and $Y$ is $(\sum_X f_X f_C)$. A similar result holds for each $C$ that is a child of $Y$ but not a child of $X$. □

**Proof of Theorem 23**

*Proof.* It suffices to show that $\Pr' = \sum_X \Pr$ when we eliminate a single variable $X$. Let $\mathcal{F}$ denote the set of CPTs for $\mathcal{M}$. Since $f_X$ is a functional CPT for $X$, we can replicate $f_X$ in $\mathcal{F}$ which yields a new CPT set (replication) $\mathcal{F}'$ that induce a same distribution as $\mathcal{F}$; see details in [48, Theorem 4]. Specifically, we pair the CPT for each child $C$ of $X$ with an extra copy of $f_X$, denoted $(f_X, f_C)$, which yields a list of pairs $(f_X, f_{C_1}), \ldots, (f_X, f_{C_k})$ where $C_1, \ldots, C_k$ are the children of $X$. Functionally eliminating $X$ from $\mathcal{F}'$ yields

$$
\sum_X \Pr = \sum_X \mathcal{F}' = \mathcal{H} \cdot (\sum_X f_X f_{C_1}) \cdots (\sum_X f_X f_{C_k})
$$

[48, Corollary 1]

$$
= \mathcal{H} \cdot f'_{C_1} \cdots f'_{C_k} = \Pr'
$$

(2)

where $\mathcal{H}$ are the CPTs in $\mathcal{F}$ that do not contain $X$ and each $f'_{C_i}$ is the CPT for child $C_i$ in $\mathcal{M}'$. □

---

[7] $X$ and $Y$ are co-parents if they have a same child.

**Proof of Theorem 24**

**Lemma 26.** *Consider a CBN $\langle G, \mathcal{F} \rangle$ and its mutilated CBN $\langle G_{\mathbf{x}}, \mathcal{F}_{\mathbf{x}} \rangle$ under $do(\mathbf{x})$. Let $W$ be a functional variable not in $\mathbf{X}$ and let $\langle G', \mathcal{F}' \rangle$ and $\langle G'_{\mathbf{x}}, \mathcal{F}'_{\mathbf{x}} \rangle$ be the results of functionally eliminating $W$ from $\langle G, \mathcal{F} \rangle$ and $\langle G_{\mathbf{x}}, \mathcal{F}_{\mathbf{x}} \rangle$, respectively. Then $\langle G'_{\mathbf{x}}, \mathcal{F}'_{\mathbf{x}} \rangle$ is the mutilated CBN for $\langle G', \mathcal{F}' \rangle$.*

*Proof.* First observe that the children of $W$ in $G$ and $G_{\mathbf{x}}$ can only differ by the variables in $\mathbf{X}$. Let $\mathbf{C}_1$ be the children of $W$ in both $G$ and $G_{\mathbf{x}}$ and let $\mathbf{C}_2$ be the children of $W$ in $G$ but not in $G_{\mathbf{x}}$. By the definition of mutilated CBN, $W$ has the same set of parents and CPT in $\langle G, \mathcal{F} \rangle$ and $\langle G_{\mathbf{x}}, \mathcal{F}_{\mathbf{x}} \rangle$. Similarly, each child $C \in \mathbf{C}_1$ has the same set of parents and CPT in $\langle G, \mathcal{F} \rangle$ and $\langle G_{\mathbf{x}}, \mathcal{F}_{\mathbf{x}} \rangle$. Hence, eliminating $W$ yields the same set of parents and CPT for each $C \in \mathbf{C}$ in $\langle G', \mathcal{F}' \rangle$ and $\langle G'_{\mathbf{x}}, \mathcal{F}'_{\mathbf{x}} \rangle$. We next consider the set of parents and CPT for each child $C \in \mathbf{C}_2$. Since $W$ is not a parent of $C$ in $\langle G_{\mathbf{x}}, \mathcal{F}_{\mathbf{x}} \rangle$, variable $C$ has the same set of parents and CPT in $\langle G'_{\mathbf{x}}, \mathcal{F}'_{\mathbf{x}} \rangle$, The exactly same set of parents (empty) and CPT will be assigned to $C$ in the mutilated CBN for $\langle G', \mathcal{F}' \rangle$. $\qquad\square$

*Proof.* (Theorem 24) Consider a CBN $\langle G, \mathcal{F} \rangle$ and its mutilated CBN $\langle G_{\mathbf{x}}, \mathcal{F}_{\mathbf{x}} \rangle$. Let $\mathrm{Pr}$ and $\mathrm{Pr}_{\mathbf{x}}$ be the distributions induced by $\mathcal{F}$ and $\mathcal{F}_{\mathbf{x}}$ over variables $\mathbf{V}$, respectively. By Lemma 26, we can eliminate each $W \in \mathbf{W}$ inductively from $\langle G, \mathcal{F} \rangle$ and $\langle G_{\mathbf{x}}, \mathcal{F}_{\mathbf{x}} \rangle$ and obtain $\langle G', \mathcal{F}' \rangle$ and its mutilated CBN $\langle G'_{\mathbf{x}}, \mathcal{F}'_{\mathbf{x}} \rangle$. By Theorem 23, the distribution induced by $\mathcal{F}'_{\mathbf{x}}$ is exactly $\sum_{\mathbf{W}} \mathrm{Pr}_{\mathbf{x}}(\mathbf{V})$. $\qquad\square$

**Proof of Proposition 10**

*Proof.* First note that the extended set $\mathbf{Z}'$ contains $\mathbf{Z}$ and all variable that are functionally determined by $\mathbf{Z}$. Consider any path $\mathcal{P}$ between some $X \in \mathbf{X}$ and $Y \in \mathbf{Y}$. We show that $\mathcal{P}$ is blocked by $\mathbf{Z}'$ iff it is blocked by $\mathbf{Z}$ according to the definition in [42]. We first show the if-part. Suppose there is a convergent valve[8] for variable $W$ that is closed when conditioned on $\mathbf{Z}$, then the valve is still closed when conditioned on $\mathbf{Z}'$ unless the parents of $W$ are in $Z'$. However, the path $\mathcal{P}$ will be blocked in the latter case since the parents of $W$ must have sequential/divergent valves. Suppose there is a sequential/divergent valve that is closed when conditioned on $\mathbf{Z}$ according to [42], then $W$ must be in $\mathbf{Z}'$ since it is functionally determined by $\mathbf{Z}$. Hence, the valve is also closed when conditioned on $\mathbf{Z}'$.

We next show the only-if part. Suppose a convergent valve for variable $W$ is closed when conditioned on $\mathbf{Z}'$, then none of $\mathbf{Z}$ is a descendent of $W$ since $\mathbf{Z}'$ is a superset of $\mathbf{Z}$. Suppose a sequential/divergent valve for variable $W$ is closed when conditioned on $\mathbf{Z}'$, then $W$ is functionally determined by $\mathbf{Z}$ by the construction of $\mathbf{Z}'$. Thus, the valve is closed in [42]. $\qquad\square$

**Proof of Theorem 11**

*Proof.* By induction, it suffices to show that $\mathbf{X}$ and $\mathbf{Y}$ are D-separated by $\mathbf{Z}$ in $\langle G, \mathbf{W} \rangle$ iff they are D-separated by $\mathbf{Z}$ in $\langle G', \mathbf{W}' \rangle$, where $G'$ is the result of functionally eliminating a single variable $T \in \mathbf{W}$ from $G'$ and $\mathbf{W}' = \mathbf{W} \setminus \{T\}$. We first show the contrapositive of the if-part. Suppose $\mathbf{X}$ and $\mathbf{Y}$ are not D-separated by $\mathbf{Z}$ in $\langle G, \mathbf{W} \rangle$, by the completeness of D-separation, there exists a parameterization $\mathcal{F}$ on $G$ such that $(\mathbf{X} \not\perp\!\!\!\perp \mathbf{Y} | \mathbf{Z})_{\mathcal{F}}$. If we eliminate $T$ from the CBN $\langle G, \mathcal{F} \rangle$, we obtain another CBN $\langle G', \mathcal{F}' \rangle$ where $\mathcal{F}'$ is the parameterization for $G'$. By Theorem 23, the marginal probabilities are preserved for the variables in $G'$, which include $\mathbf{X}, \mathbf{Y}, \mathbf{Z}$. Hence, $(\mathbf{X} \not\perp\!\!\!\perp \mathbf{Y} | \mathbf{Z})_{\mathcal{F}'}$ and $\mathbf{X}$ and $\mathbf{Y}$ are not D-separated by $\mathbf{Z}$ in $\langle G', \mathbf{W}' \rangle$.

Next consider the contrapositive of the only-if part. Suppose $\mathbf{X}$ and $\mathbf{Y}$ are not D-separated by $\mathbf{Z}$ in $\langle G', \mathbf{W}' \rangle$, then there exists a parameterization $\mathcal{F}'$ of $G'$ such that $(\mathbf{X} \not\perp\!\!\!\perp \mathbf{Y} | \mathbf{Z})_{\mathcal{F}'}$ by the completeness of D-separation. We construct a parameterization $\mathcal{F}$ for $G$ such that $\mathcal{F}'$ is the parameterization of $G'$ which results from eliminating $T$ from the CBN $\langle G, \mathcal{F} \rangle$. This is sufficient to show that $\mathbf{X}$ and $\mathbf{Y}$ are not D-separated by $\mathbf{Z}$ in $\langle G, \mathbf{W} \rangle$ since the marginals are preserved by Theorem 23.

**Construction Method** Let $\mathbf{P}_T$ and $\mathbf{C}_T$ denote the parents and children of $T$ in $G$. Our construction assumes that the cardinality of $T$ is the number of instantiations for its parents $\mathbf{P}_T$. That is, there is a one-to-one correspondence between the states of $T$ and the instantiations of $\mathbf{P}_T$, and we use $\alpha(t)$ to denote the instantiation $\mathbf{p}_T$ corresponding to state $t$. The functional CPT for $T$ is assigned as $f_T(t|\mathbf{p}_T) = 1$ if $\alpha(t) = \mathbf{p}_T$ and $f_T(t|\mathbf{p}_T) = 0$ otherwise for each instantiation $\mathbf{p}_T$ of $\mathbf{P}_T$. Now consider each child $C \in \mathbf{C}_T$ that has parents $\mathbf{P}_C$ (excluding $T$) and $T$ in $G$. It immediately follows

---

[8]See [40, Ch. 4] for more details on convergent, divergent and sequential valves.

from Definition 9 that $C$ has parents $\mathbf{P}_T \cup \mathbf{P}_C$ in $G'$. We next construct the CPT $f_C$ in $\mathcal{F}$ based on its CPT $f'_C$ in $\mathcal{F}'$. Consider each parent instantiation $(t, \mathbf{p}_C)$ where $t$ is a state of $T$ and $\mathbf{p}_C$ is an instantiation of $\mathbf{P}_C$. If $\alpha(t)$ is consistent with $\mathbf{p}_C$, assign $f_C(c|t, \mathbf{p}_C) = f'_C(c|\alpha(t), \mathbf{p}_C)$ for each state $c$.[9] Otherwise, assign any functional distribution for $f_C(C|t, \mathbf{p}_C)$. The construction ensures that the constructed CPT $f_T$ for $T$ is functional, and that the functional dependencies among other variables are preserved. In particular, for each child $C$ of $T$, the constructed CPT $f_C$ is functional iff $f'_C$ is functional. This construction method will be reused later in other proofs.

We now just need to show that CBN $\langle G', \mathcal{F}' \rangle$ is the result of eliminating $T$ from the (constructed) CBN $\langle G, \mathcal{F} \rangle$. In particular, we need to check that the CPT for each child $C \in \mathbf{C}_T$ in $\mathcal{F}'$ is correctly computed from the constructed CPTs in $\mathcal{F}$. For each instantiation $(\mathbf{p}_T, \mathbf{p}_C)$ and state $c$ of $C$,

$$f'_C(c|\mathbf{p}_T, \mathbf{p}_C) = f_C(c|t^*, \mathbf{p}_C) = f_T(t^*|\mathbf{p}_T) f_C(c|t^*, \mathbf{p}_C)$$
$$= \sum_t f_T(t|\mathbf{p}_T) f_C(c|t, \mathbf{p}_C)$$

where $t^*$ is the state of $T$ such that $\alpha(t^*) = \mathbf{p}_T$. $\qquad\square$

**Proof of Theorem 13**

*Proof.* We prove the theorem by induction. It suffices to show the following statement: for each causal graph $G$ with observed variables $\mathbf{V}$ and functional variables $\mathbf{W}$, the causal effect $\Pr_{\mathbf{x}}(\mathbf{Y})$ is F-identifiable wrt $\langle G, \mathbf{V}, \mathcal{C}_{\mathbf{V}}, \mathbf{W} \rangle$ iff it is F-identifiable wrt $\langle G', \mathbf{V}, \mathcal{C}_{\mathbf{V}}, \mathbf{W}' \rangle$ where $G'$ is the result of functionally eliminating some hidden functional variable $T \in \mathbf{W}$ and $\mathbf{W}' = \mathbf{W} \setminus \{T\}$.

We first show the contrapositive of the if-part. Suppose $\Pr_{\mathbf{x}}(\mathbf{Y})$ is not F-identifiable wrt $\langle G, \mathbf{V}, \mathcal{C}_{\mathbf{V}}, \mathbf{W} \rangle$, there exist two CBNs $\langle G, \mathcal{F}_1 \rangle$ and $\langle G, \mathcal{F}_2 \rangle$ which induce distributions $\Pr_1, \Pr_2$ such that $\Pr_1(\mathbf{V}) = \Pr_2(\mathbf{V})$ but $\Pr_{1\mathbf{x}}(\mathbf{Y}) \neq \Pr_{2\mathbf{x}}(\mathbf{Y})$. Let $\langle G', \mathcal{F}'_1 \rangle$ and $\langle G', \mathcal{F}'_2 \rangle$ be the results of eliminating $T \notin \mathbf{V}$ from $\langle G, \mathcal{F}_1 \rangle$ and $\langle G, \mathcal{F}_2 \rangle$, the two CBNs attain the same marginal distribution on $\mathbf{V}$ but different causal effects by Theorem 23 and Theorem 24. Hence, $\Pr_{\mathbf{x}}(\mathbf{Y})$ is not F-identifiable wrt $\langle G', \mathbf{V}, \mathcal{C}_{\mathbf{V}}, \mathbf{W}' \rangle$ either.

We next show the contrapositive of the only-if part. Suppose $\Pr_{\mathbf{x}}(\mathbf{Y})$ is not F-identifiable wrt $\langle G', \mathbf{V}, \mathcal{C}_{\mathbf{V}}, \mathbf{W}' \rangle$, there exist two CBNs $\langle G', \mathcal{F}'_1 \rangle$ and $\langle G', \mathcal{F}'_2 \rangle$ which induce distributions $\Pr'_1, \Pr'_2$ such that $\Pr'_1(\mathbf{V}) = \Pr'_2(\mathbf{V})$ but $\Pr'_{1\mathbf{x}}(\mathbf{Y}) \neq \Pr'_{2\mathbf{x}}(\mathbf{Y})$. We can obtain $\langle G, \mathcal{F}_1 \rangle$ and $\langle G, \mathcal{F}_2 \rangle$ by considering again the construction method in Theorem 11 where we assign a one-to-one mapping for $T$ and adopt the CPTs from $\mathcal{F}'_1$ and $\mathcal{F}'_2$ for the children of $T$. This way, $\langle G', \mathcal{F}'_1 \rangle$ and $\langle G', \mathcal{F}'_2 \rangle$ become the results of eliminating $T$ from the constructed $\langle G, \mathcal{F}_1 \rangle$ and $\langle G, \mathcal{F}_2 \rangle$. Since $T \notin \mathbf{V}$, $\Pr_1(\mathbf{V}) = \Pr'_1(\mathbf{V}) = \Pr'_2(\mathbf{V}) = \Pr_2(\mathbf{V})$ by Theorem 23, and $\Pr_{1\mathbf{x}}(\mathbf{Y}) = \Pr'_{1\mathbf{x}}(\mathbf{Y}) \neq \Pr'_{2\mathbf{x}}(\mathbf{Y}) = \Pr_{2\mathbf{x}}(\mathbf{Y})$ by Theorem 24. Hence, $\Pr_{\mathbf{x}}(\mathbf{Y})$ is not F-identifiable wrt $\langle G, \mathbf{V}, \mathcal{C}_{\mathbf{V}}, \mathbf{W} \rangle$ either. $\qquad\square$

**Proof of Theorem 15**

*Proof.* Since we only functionally eliminate variables that have observed parents, it is guaranteed that each $Z \in \mathbf{Z}$ has observed parents when it is eliminated. By induction, it suffices to show that $\Pr_{\mathbf{x}}(\mathbf{Y})$ is F-identifiable wrt $\langle G, \mathbf{V}, \mathcal{C}_{\mathbf{V}}, \mathbf{W} \rangle$ iff it is F-identifiable wrt $\langle G', \mathbf{V}', \mathcal{C}_{\mathbf{V}}, \mathbf{W}' \rangle$ where $G'$ is the result of eliminating a single functional variable $Z \in \mathbf{W}$ with observed parents from $G$, $\mathbf{V}' = \mathbf{V} \setminus \{Z\}$, and $\mathbf{W}' = \mathbf{W} \setminus \{Z\}$.

We first show the contrapositive of the if-part. Suppose $\Pr_{\mathbf{x}}(\mathbf{Y})$ is not F-identifiable wrt $\langle G, \mathbf{V}, \mathcal{C}_{\mathbf{V}}, \mathbf{W} \rangle$, there exist two CBNs $\langle G, \mathcal{F}_1 \rangle$ and $\langle G, \mathcal{F}_2 \rangle$ which induce distributions $\Pr_1, \Pr_2$ where $\Pr_1(\mathbf{V}) = \Pr_2(\mathbf{V})$ but $\Pr_{1\mathbf{x}}(\mathbf{Y}) \neq \Pr_{2\mathbf{x}}(\mathbf{Y})$. Let $\langle G', \mathcal{F}'_1 \rangle$ and $\langle G', \mathcal{F}'_2 \rangle$ be the results of eliminating $Z$ from $\langle G, \mathcal{F}_1 \rangle$ and $\langle G, \mathcal{F}_2 \rangle$, the two CBNs induce the same marginal distribution $\Pr'_1(\mathbf{V}') = \Pr'_2(\mathbf{V}')$ by Theorem 23 but different causal effects $\Pr'_{1\mathbf{x}}(\mathbf{Y}) \neq \Pr'_{2\mathbf{x}}(\mathbf{Y})$ by Theorem 24. Hence, $\Pr_{\mathbf{x}}(\mathbf{Y})$ is not F-identifiable wrt $\langle G', \mathbf{V}', \mathcal{C}_{\mathbf{V}}, \mathbf{W}' \rangle$.

We now consdier the ctrapositive of the only-if part. Suppose $\Pr_{\mathbf{x}}(\mathbf{Y})$ is not F-identifiable wrt $\langle G', \mathbf{V}', \mathcal{C}_{\mathbf{V}}, \mathbf{W}' \rangle$, then there exist two CBNs $\langle G', \mathcal{F}'_1 \rangle$ and $\langle G', \mathcal{F}'_2 \rangle$ which induce distributions $\Pr'_1, \Pr'_2$ such that $\Pr'_1(\mathbf{V}') = \Pr'_2(\mathbf{V}')$ but $\Pr'_{1\mathbf{x}}(\mathbf{Y}) \neq \Pr'_{2\mathbf{x}}(\mathbf{Y})$. We again consider the construction method from the proof of Theorem 11 which produces two CBNs $\langle G, \mathcal{F}_1 \rangle$ and $\langle G, \mathcal{F}_2 \rangle$.

---

[9] For clarity, we use the notation | to separate a variable and its parents in a CPT.

Moreover, $\langle G', \mathcal{F}_1' \rangle$ and $\langle G', \mathcal{F}_2' \rangle$ are the result of eliminating $Z$ from $\langle G, \mathcal{F}_1 \rangle$ and $\langle G, \mathcal{F}_2 \rangle$. It is guaranteed that the two constructed CBNs produce different causal effects $\mathrm{Pr}_{1\mathbf{x}}(\mathbf{Y}) = \mathrm{Pr}_{1\mathbf{x}}'(\mathbf{Y}) \neq \mathrm{Pr}_{2\mathbf{x}}'(\mathbf{Y}) = \mathrm{Pr}_{2\mathbf{x}}(\mathbf{Y})$ by Theorem 24. We need to show that $\langle G, \mathcal{F}_1 \rangle$ and $\langle G, \mathcal{F}_2 \rangle$ induce a same distribution over variables $\mathbf{V} = \mathbf{V}' \cup \{Z\}$. Consider any instantiation $(\mathbf{v}', z)$ of $\mathbf{V}$. Since $\mathcal{F}_1$ and $\mathcal{F}_2$ assign the same one-to-one mapping $\alpha$ between $Z$ and its parents in $G$, it is guaranteed that the probabilities $\mathrm{Pr}_1(\mathbf{v}', z) = \mathrm{Pr}_2(\mathbf{v}', z) = 0$ except for the single state $z^*$ where $\alpha(z^*) = \mathbf{p}$ where $\mathbf{p}$ is the parent instantiation of $Z$ consistent with $\mathbf{v}'$. By construction, $\mathrm{Pr}_1(\mathbf{v}', z^*, \mathbf{u}) = \mathrm{Pr}_1'(\mathbf{v}', \mathbf{u})$ for every instantiation $\mathbf{u}$ of hidden variables $\mathbf{U}$. Similarly, $\mathrm{Pr}_2(\mathbf{v}', z^*, \mathbf{u}) = \mathrm{Pr}_2'(\mathbf{v}', \mathbf{u})$ for every instantiation $(\mathbf{v}', z, \mathbf{u})$. It then follows that $\mathrm{Pr}_1(\mathbf{v}', z^*) = \sum_{\mathbf{u}} \mathrm{Pr}_1(\mathbf{v}', z^*, \mathbf{u}) = \sum_{\mathbf{u}} \mathrm{Pr}_1'(\mathbf{v}', \mathbf{u}) = \mathrm{Pr}_1'(\mathbf{v}')$ $= \mathrm{Pr}_2'(\mathbf{v}') = \sum_{\mathbf{u}} \mathrm{Pr}_2'(\mathbf{v}', \mathbf{u}) = \sum_{\mathbf{u}} \mathrm{Pr}_2(\mathbf{v}', z^*, \mathbf{u}) = \mathrm{Pr}_2(\mathbf{v}', z^*)$. This means $\mathrm{Pr}_{\mathbf{x}}(\mathbf{Y})$ is not F-identifiable wrt $\langle G, \mathbf{V}, \mathcal{C}_{\mathbf{V}}, \mathbf{W} \rangle$ either. $\qquad \square$

**Proof of Theorem 17**

**Lemma 27.** *Let $G$ be a causal graph, $\mathbf{V}$ be its observed variables and $\mathbf{W}$ be its functional variables. Let $Z$ be a non-descendant of $\mathbf{W}$ that has at least one hidden parent, then a causal effect is F-identifiable wrt $\langle G, \mathbf{V}, \mathcal{C}_{\mathbf{V}}, \mathbf{W} \rangle$ iff it is F-identifiable wrt $\langle G, \mathbf{V}, \mathcal{C}_{\mathbf{V}}, \mathbf{W} \cup \{Z\} \rangle$.*

*Proof.* Let $\mathbf{W}'$ denote the set $\mathbf{W} \cup \{Z\}$. The only-if part holds immediately by the fact that every distribution that can be possibly induced from $\langle G, \mathbf{W}' \rangle$ can also be induced from $\langle G, \mathbf{W} \rangle$. We next consider the contrapositive of the if part. Suppose a causal effect is not F-identifiable wrt $\langle G, \mathbf{V}, \mathcal{C}_{\mathbf{V}}, \mathbf{W} \rangle$, then there exist two CBNs $\langle G, \mathcal{F}_1 \rangle$ and $\langle G, \mathcal{F}_2 \rangle$ which induce distributions $\mathrm{Pr}_1, \mathrm{Pr}_2$ such that $\mathrm{Pr}_1(\mathbf{V}) = \mathrm{Pr}_2(\mathbf{V})$ but $\mathrm{Pr}_{1\mathbf{x}}(\mathbf{Y}) \neq \mathrm{Pr}_{2\mathbf{x}}(\mathbf{Y})$. We next construct $\langle G, \mathcal{F}_1'' \rangle$ and $\langle G, \mathcal{F}_2'' \rangle$ which constitute an example of unidentifiability wrt $\langle G, \mathbf{V}, \mathcal{C}_{\mathbf{V}}, \mathbf{W}' \rangle$. In particular, the CPTs for $Z$ need to be functional in the constructed CBNs.

WLG, we show the construction of $\langle G, \mathcal{F}_1'' \rangle$ from $\langle G, \mathcal{F}_1 \rangle$ which involves two steps (the construction of $\langle G, \mathcal{F}_2'' \rangle$ from $\langle G, \mathcal{F}_2 \rangle$ will follow a same procedure). Let $\mathbf{P}$ be the parents of $Z$. The first step constructs a CBN $\langle G', \mathcal{F}_1' \rangle$ based on the known method that transforms any (non-functional) CPT into a functional CPT. This is done by adding an auxiliary hidden root parent $U$ for $Z$ whose states correspond to the possible functions between $\mathbf{P}$ and $Z$. The CPTs for $U$ and $Z$ are assigned accordingly such that $f_Z = \sum_U f_U' f_Z'$ where $f_U'$ and $f_Z'$ are the constructed CPTs in $\mathcal{F}_1'$.[10] It follows that $\mathcal{F}_1$ and $\mathcal{F}_1'$ induce the same distribution over $\mathbf{V}$ since $\mathcal{F}_1 = \sum_U \mathcal{F}_1'$. The causal effect is also preserved since summing out $U$ is independent of other CPTs in the mutilated CBN for $\langle G', \mathcal{F}_1' \rangle$.

Our second step involves converting the CBN $\langle G', \mathcal{F}_1' \rangle$ (constructed from the first step) into the CBN $\langle G, \mathcal{F}_1'' \rangle$ over the original graph $G$. Let $T \in \mathbf{P}$ be the hidden parent of $W$ in $G$. We merge the auxiliary parent $U$ and $T$ into a new variable $T'$ and substitute it for $T$ in $G$, i.e., $T'$ has the same parents and children as $T$. $T'$ is constructed as the Cartesian product of $U$ and $T$: each state of $T'$ is represented as a pair $(u, t)$ where $u$ is a state of $U$ and $t$ is a state of $T$. We are ready to assign new CPTs for $T'$ and its children. For each parent instantiation $\mathbf{p}$ of $\mathbf{P}$ and each state $(u, t)$ of $T'$, we assign the CPT for $T'$ in $\mathcal{F}''$ as $f_{T'}''((u,t)|\mathbf{p}) = f_U'(u) f_T'(t|\mathbf{p})$. Next consider each child $C$ of $T'$ that has parents $\mathbf{P}_C$ (excluding $T'$). For each instantiation $\mathbf{p}_C$ of $\mathbf{P}_C$ and each state $(u, t)$ of $T'$, we assign the CPT for $C$ in $\mathcal{F}''$ as $f_C''(c|\mathbf{p}_C, (u,t)) = f_C'(c|\mathbf{p}_C)$ for each state $c$. Note that $f_C''$ is functional iff $f_C'$ is functional. Hence, the CPTs for $\mathbf{W}$ are all functional in $\mathcal{F}''$.

We need to show that $\langle G, \mathcal{F}_1'' \rangle$ preserves the distribution on $\mathbf{V}$ and the causal effect from $\langle G', \mathcal{F}_1' \rangle$. Let $\mathbf{U}''$ be the hidden variables in $\langle G, \mathcal{F}_1'' \rangle$ and $\mathbf{U}'$ be the hidden variables in $\langle G', \mathcal{F}_1' \rangle$. The distribution on $\mathbf{V}$ is preserved since there is a one-to-one correspondence between each instantiation $(\mathbf{v}, \mathbf{u}'')$ in $\langle G, \mathcal{F}_1'' \rangle$ and each instantiation $(\mathbf{v}, \mathbf{u}')$ in $\langle G', \mathcal{F}_1' \rangle$ where the two instantiations agree on $\mathbf{v}$ and are assigned with the same probability, i.e., $\mathrm{Pr}_1''(\mathbf{v}, \mathbf{u}'') = \mathrm{Pr}_1'(\mathbf{v}, \mathbf{u}')$. Hence, $\mathrm{Pr}_1''(\mathbf{v}) = \sum_{\mathbf{u}''} \mathrm{Pr}_1''(\mathbf{v}, \mathbf{u}'') = \sum_{\mathbf{u}'} \mathrm{Pr}_1'(\mathbf{v}, \mathbf{u}') = \mathrm{Pr}_1'(\mathbf{v})$ for every instantiation $\mathbf{v}$. The preservation of causal effect can be shown similarly but on the mutilated CBNs. Thus, $\langle G, \mathcal{F}_1'' \rangle$ preserves both the distribution on $\mathbf{V}$ and the causal effect from $\langle G, \mathcal{F}_1 \rangle$. Similarly, we can construct $\langle G, \mathcal{F}_2'' \rangle$ which preserves the distribution on $\mathbf{V}$ and causal effect from $\langle G, \mathcal{F}_2 \rangle$. The two CBNs $\langle G, \mathcal{F}_1'' \rangle$ and $\langle G, \mathcal{F}_2'' \rangle$ constitute an example of unidentifiablity wrt $\langle G, \mathbf{V}, \mathcal{C}_{\mathbf{V}}, \mathbf{W}' \rangle$. $\qquad \square$

---

[10]Each state $u$ of $U$ corresponds to a function $\gamma_u$ where $\gamma_u(\mathbf{p})$ is mapped to some state of $Z$ for each instantiation $\mathbf{p}$. The variable $U$ thus has $|Z|^{|\mathbf{P}|}$ states since there are total of $|Z|^{|\mathbf{P}|}$ possible functions from $\mathbf{P}$ to $Z$. For each instantiation $(z, u, \mathbf{p})$, the functional CPT for $Z$ is defined as $f_Z'(z|u, \mathbf{p}) = 1$ if $z = \gamma_u(\mathbf{p})$, and $f_Z'(z|u, \mathbf{p}) = 0$ otherwise. The CPT for $U$ is assigned as $f_U'(u) = \prod_{\mathbf{p} \in \mathbf{P}} f_Z(\gamma_u(\mathbf{p})|\mathbf{p})$.

*Proof of Theorem 17.* We prove the theorem by induction. We first order all the functional variables in a bottom-up order. Let $W^i$ denote the $i^{th}$ functional variable in the order and $\mathbf{W}^{(i)}$ denote the functional variables that are ordered before $W^i$ (including $W^i$). It follows that we can go over each $W^i$ in the order and show that a causal effect is F-identifiable wrt $\langle G, \mathbf{V}, \mathcal{C}_{\mathbf{V}}, \mathbf{W}^{(i-1)} \rangle$ iff it is F-identifiable wrt $\langle G, \mathbf{V}, \mathcal{C}_{\mathbf{V}}, \mathbf{W}^{(i)} \rangle$ by Lemma 27. Since F-identifiability wrt $\langle G, \mathbf{V}, \mathcal{C}_{\mathbf{V}}, \mathbf{W}^{(0)} \rangle$ is equivalent to identifiability wrt $\langle G, \mathbf{V}, \mathcal{C}_{\mathbf{V}} \rangle$, we conclude that the causal effect is F-identifiable wrt $\langle G, \mathbf{V}, \mathcal{C}_{\mathbf{V}}, \mathbf{W} \rangle$ iff it is identifiabile wrt $\langle G, \mathbf{V}, \mathcal{C}_{\mathbf{V}} \rangle$. $\qquad\square$

## Proof of Theorem 18

The proof of the theorem is organized as follows. We start with a lemma (Lemma 28) that allows us to modify the CPT of a variable when the marginal probability over its parents contain zero entries. We then show a main lemma (Lemma 29) that allows us to reduce F-identifiability to identifiability when all functional variables are observed or has a hidden parent. We finally prove the theorem based on the main lemma and previous theorems.

**Lemma 28.** *Consider two CBNs that have a same causal graph and induce the distributions* $\Pr_1$ *and* $\Pr_2$*. Suppose the CPTs of the two CBNs only differ by* $f_X(X, \mathbf{P})$*. Then* $\Pr_1(\mathbf{p}) = 0$ *iff* $\Pr_2(\mathbf{p}) = 0$ *for all instantiations* $\mathbf{p}$ *of* $\mathbf{P}$*.*

*Proof.* Let $f_Y$ and $g_Y$ denote the CPT for $Y$ in the first and second CBN. Let $\mathcal{An}(\mathbf{P})$ be the ancestors of variables in $\mathbf{P}$ (including $\mathbf{P}$). If we eliminate all variables other than $\mathcal{An}(\mathbf{P})$, then we obtain the factor set $\mathcal{F}_1 = \prod_{Y \in \mathcal{An}(\mathbf{P})} f_Y$ for the first CBN and $\mathcal{F}_2 = \prod_{Y \in \mathcal{An}(\mathbf{P})} g_Y$ for the second CBN. Since all CPTs are the same for variables in $\mathcal{An}(\mathbf{P})$, it is guaranteed that $\mathcal{F}_1 = \mathcal{F}_2$. If we further eliminate variables other than $\mathbf{P}$ from $\mathcal{F}_1$ and $\mathcal{F}_2$, we obtain the marginal distributions $\Pr_1(\mathbf{P}) = \overline{\overline{\sum}}_{\mathbf{P}} \mathcal{F}_1$ and $\Pr_2(\mathbf{P}) = \overline{\overline{\sum}}_{\mathbf{P}} \mathcal{F}_2$, where $\overline{\overline{\sum}}_{\mathbf{P}}$ denotes the projection operation that sums out variables other than $\mathbf{P}$ from a factor. Hence, $\Pr_1(\mathbf{P}) = \Pr_2(\mathbf{P})$ which concludes the proof. $\qquad\square$

**Lemma 29.** *If a causal effect is F-identifiable wrt* $\langle G, \mathbf{V}, \mathcal{C}_{\mathbf{V}}, \mathbf{W} \rangle$ *but is not identifiable* $\langle G, \mathbf{V}, \mathcal{C}_{\mathbf{V}} \rangle$*, then there must exist at least one functional variable that is hidden and whose parents are all observed.*

*Proof.* The lemma is the same as saying that if every functional variable is observed or having a hidden parent, then F-identifiability is equivalent to identifiability. We go over each functional variable $W_i \in \mathbf{W}$ in a bottom-up order $\Pi$ and show the following inductive statement: a causal effect $\Pr_x(\mathbf{Y})$ is F-identifiable wrt $\langle G, \mathbf{V}, \mathcal{C}_{\mathbf{V}}, \mathbf{W}^{(i)} \rangle$ iff it is F-identifiable wrt $\langle G, \mathbf{V}, \mathcal{C}_{\mathbf{V}}, \mathbf{W}^{(i-1)} \rangle$, where $\mathbf{W}^{(i)}$ is a subset of variables in $\mathbf{W}$ that are ordered before $W_i$ (and including $W_i$) in $\Pi$. Note that $\mathbf{W}^{(0)} = \emptyset$ and F-identifiability wrt $\langle G, \mathbf{V}, \mathcal{C}_{\mathbf{V}}, \mathbf{W}^{(0)} \rangle$ collapses into identifiability wrt $\langle G, \mathbf{V}, \mathcal{C}_{\mathbf{V}} \rangle$.

The if-part follows from the definitions of identifiability and F-identifiability. We next consider the contrapositive of the only-if part. Let $Z$ be the functional variable in $\mathbf{W}$ that is considered in the current inductive step. Let $\langle G, \mathcal{F}_1 \rangle$ and $\langle G, \mathcal{F}_2 \rangle$ be the two CBNs inducing distributions $\Pr_1$ and $\Pr_2$ that constitute the unidentifiability, i.e., $\Pr_1(\mathbf{V}) = \Pr_2(\mathbf{V})$ and $\Pr_{1x}(\mathbf{Y}) \neq \Pr_{2x}(\mathbf{Y})$. Our goal is to construct two CBNs $\langle G, \mathcal{F}_1''' \rangle$ and $\langle G, \mathcal{F}_2''' \rangle$, which induce distributions $\Pr_1'''$, $\Pr_2'''$ and contain functional CPTs for $Z$, such that $\Pr_1'''(\mathbf{V}) = \Pr_2'''(\mathbf{V})$ and $\Pr_{1x}'''(\mathbf{Y}) \neq \Pr_{2x}'''(\mathbf{Y})$. Suppose $Z$ has a hidden parent, we directly employ Lemma 27 to construct the two CBNs. We next consider the case when $Z$ is observed and has observed parents. By default, we use $f_Z$ to $g_Z$ to denote the CPT for $Z$ in $\mathcal{F}_1$ and $\mathcal{F}_2$.

The following three steps are considered to construct an instance of unidentifiability.

**First Step:** we construct $\langle G, \mathcal{F}_1' \rangle$ and $\langle G, \mathcal{F}_2' \rangle$ by modifying the CPTs for $Z$. Let $\mathbf{P}$ be the parents of $Z$ in $G$. For each instantiation $\mathbf{p}$ of $\mathbf{P}$ where $\Pr_1(\mathbf{p}) = \Pr_2(\mathbf{p}) = 0$, we modify the entries $f_Z'(Z|\mathbf{p})$ and $g_Z'(Z|\mathbf{p})$ for the CPTs $f_Z'$ and $g_Z'$ in $\mathcal{F}_1'$ and $\mathcal{F}_2'$ as follows. Since $\Pr_{1x}(\mathbf{Y}) \neq \Pr_{2x}(\mathbf{Y})$, there exists an instantiation $\mathbf{y}$ such that $\Pr_{1x}(\mathbf{y}) \neq \Pr_{2x}(\mathbf{y})$. WLG, assume $\Pr_{1x}(\mathbf{y}) > \Pr_{2x}(\mathbf{y})$. Since $\Pr_{1x}(\mathbf{y})$ is computed as the marginal probability of $\mathbf{y}$ in the mutilated CBN for $\langle G, \mathcal{F}_1 \rangle$, it can be expressed in the form of *network polynomial* as shown in [49, 40]. If we treat the CPT entries of $f_Z(Z|\mathbf{p})$ as unknown, then we can write $\Pr_{1x}(\mathbf{y})$ as follows

$$\mathrm{Pr}_{1x}(\mathbf{y}) = \alpha_0 + \alpha_1 f_Z(z_1|\mathbf{p}) + \cdots + \alpha_k f_Z(z_k|\mathbf{p})$$

where $\alpha_0, \alpha_1, \ldots, \alpha_k$ are constants and $z_1, \ldots, z_k$ are the states of variable $Z$. Similarly, we can write $\mathrm{Pr}_{2x}(\mathbf{y})$ as follows

$$\mathrm{Pr}_{2x}(\mathbf{y}) = \beta_0 + \beta_1 g_Z(z_1|\mathbf{p}) + \cdots + \beta_k g_Z(z_k|\mathbf{p})$$

Let $\alpha_i$ be the *maximum* value among $\alpha_1, \ldots, \alpha_k$ and $\beta_j$ be the *minimum* value among $\beta_1, \ldots, \beta_k$, our construction method assigns $f'_Z(z_i|\mathbf{p}) = 1$ and $g'_Z(z_j|\mathbf{p}) = 1$. By construction, it is guaranteed that $\mathrm{Pr}'_{1x}(\mathbf{y}) - \mathrm{Pr}'_{2x}(\mathbf{y}) \geq \mathrm{Pr}_{1x}(\mathbf{y}) - \mathrm{Pr}_{2x}(\mathbf{y}) > 0$ where $\mathrm{Pr}'_{1x}(\mathbf{y})$, $\mathrm{Pr}'_{2x}(\mathbf{y})$ denote the causal effects under the updated CPTs $f'_Z(Z|\mathbf{p})$ and $g'_Z(Z|\mathbf{p})$. We repeat the above procedure for all such $\mathbf{p}$ where $\mathrm{Pr}_1(\mathbf{p}) = 0$, which yields the new CBNs $\langle G, \mathcal{F}'_1 \rangle$ and $\langle G, \mathcal{F}'_2 \rangle$ in which $f'_Z(Z|\mathbf{p})$ and $g'_Z(Z|\mathbf{p})$ are functional whenever $\mathrm{Pr}_1(\mathbf{p}) = 0$. We next show that $\mathcal{F}'_1$ and $\mathcal{F}'_2$ (with the updated $f'_Z$ and $g'_Z$) constitute an example of unidentifiability. $\mathrm{Pr}'_{1x}(\mathbf{Y}) \neq \mathrm{Pr}'_{2x}(\mathbf{Y})$ since $\mathrm{Pr}'_{1x}(\mathbf{y}) > \mathrm{Pr}'_{2x}(\mathbf{y})$ for the particular instantiation $\mathbf{y}$. We are left to show that the distributions $\mathrm{Pr}'_1$ and $\mathrm{Pr}'_2$ induced by $\langle G, \mathcal{F}'_1 \rangle$ and $\langle G, \mathcal{F}'_2 \rangle$ are the same over the observed variables $\mathbf{V}$. Consider each instantiation $\mathbf{v}$ of $\mathbf{V}$ and $\mathbf{p}$ of $\mathbf{P}$ where $\mathbf{p}$ is consistent with $\mathbf{v}$. If $\mathrm{Pr}_1(\mathbf{p}) = 0$, then $\mathrm{Pr}'_1(\mathbf{p}) = \mathrm{Pr}'_2(\mathbf{p}) = 0$ by Lemma 28 and thus $\mathrm{Pr}'_1(\mathbf{v}) = \mathrm{Pr}'_2(\mathbf{v}) = 0$. Otherwise, $\mathrm{Pr}'_1(\mathbf{v}) = \mathrm{Pr}_1(\mathbf{v}) = \mathrm{Pr}_2(\mathbf{v}) = \mathrm{Pr}'_2(\mathbf{v})$ since none of the CPT entries consistent with $\mathbf{v}$ were modified.

**Second Step:** We construct $\langle G'', \mathcal{F}''_1 \rangle$, $\langle G'', \mathcal{F}''_2 \rangle$ from $\langle G, \mathcal{F}'_1 \rangle$, $\langle G, \mathcal{F}'_2 \rangle$ by introducing an auxiliary root parent for $Z$ and assigning a functional CPT for $Z$. We add a root variable, denoted $R$, to be an auxiliary parent of $W$ which specifies all possible functions from $\mathbf{P}^*$ to $Z$ where $\mathbf{P}^*$ contains all instantiations of $\mathbf{p}^*$ where $\mathrm{Pr}'_1(\mathbf{p}^*) = \mathrm{Pr}'_2(\mathbf{p}^*) > 0$. Each state $r$ of $R$ corresponds to a function $\varphi_r$ where $\varphi_r(\mathbf{p}^*)$ is mapped to some state of $Z$ for each instantiation $\mathbf{p}^*$. The variable $R$ thus has $|Z|^{|\mathbf{P}^*|}$ states since there are total of $|Z|^{|\mathbf{P}^*|}$ possible functions from $\mathbf{P}^*$ to $Z$. For each instantiation $(z, r, \mathbf{p})$, if $\mathbf{p} \in \mathbf{P}^*$, we define $f''_Z(z|r, \mathbf{p}) = 1$ if $z = \varphi_r(\mathbf{p})$, and $f''_Z(z|r, \mathbf{p}) = 0$ otherwise. If $\mathbf{p} \notin \mathbf{P}^*$, we define $f''_Z(z|r, \mathbf{p}) = f'_Z(z|\mathbf{p})$. The CPT for $R$ is assigned as $f''_R(r) = \prod_{\mathbf{p} \in \mathbf{P}^*} f'_Z(\varphi_r(\mathbf{p})|\mathbf{p})$. Moreover, we remove all the states $r$ of $R$ where $f''_R(r) = 0$, which ensures $f''_R(r) > 0$ for all remaining $r$. We assign the CPT $g''_Z$ in $\mathcal{F}''_2$ in a similar way. Note that $f''_R = g''_R$ since $f'_Z(\varphi_r(\mathbf{p})|\mathbf{p}) = \mathrm{Pr}'_1(\varphi_r(\mathbf{p})|\mathbf{p}) = \mathrm{Pr}'_2(\varphi_r(\mathbf{p})|\mathbf{p}) = g'_Z(\varphi_r(\mathbf{p})|\mathbf{p})$ for each $\mathbf{p} \in \mathbf{P}^*$ (where $\mathrm{Pr}'_1(\mathbf{p}) = \mathrm{Pr}'_2(\mathbf{p}) > 0$).

We next show that $\langle G'', \mathcal{F}''_1 \rangle$ and $\langle G'', \mathcal{F}''_2 \rangle$ constitute the unidentifiability. One key observation is that $f'_T(t|\mathbf{p}) = \sum_r f''_R(r) f''_T(t|\mathbf{p}, r)$ and $g'_T(t|\mathbf{p}) = \sum_r g''_R(r) g''_T(t|\mathbf{p}, r)$. Consider each instantiation $(\mathbf{v}, r)$ which contains the instantiation $\mathbf{p}$ of $\mathbf{P}$ and the state $z$ of $Z$. Suppose $\mathrm{Pr}'_1(\mathbf{p}) = 0$, then $\mathrm{Pr}''_1(\mathbf{p}) = 0$ since the marginal over $\mathbf{V}$ is preserved in $\mathrm{Pr}''$. Hence, $\mathrm{Pr}''_1(\mathbf{v}, r) = \mathrm{Pr}''_2(\mathbf{v}, r) = 0$. Suppose $\mathrm{Pr}'_1(\mathbf{p}) \neq 0$, then $\mathrm{Pr}''_1(\mathbf{v}, r) = (\prod_{V \in \mathbf{V} \setminus \{Z\}} f'_V) \cdot f''_R(r)$ when $z = \varphi_r(\mathbf{p})$, and $\mathrm{Pr}''_1(\mathbf{v}, r) = 0$ otherwise. Similarly, $\mathrm{Pr}''_2(\mathbf{v}, r) = (\prod_{V \in \mathbf{V} \setminus \{Z\}} g'_V) \cdot g''_R(r)$ when $z = \varphi_r(\mathbf{p})$, and $\mathrm{Pr}''_2(\mathbf{v}, r) = 0$ otherwise. In both cases, $\mathrm{Pr}''_1(\mathbf{v}, r) = \mathrm{Pr}''_2(\mathbf{v}, r)$ since $\mathcal{F}''_1$ and $\mathcal{F}''_2$ assign a same function $\varphi_r$ for each state $r$ of $R$, and $f'_V = g'_V$ for all $V \in \mathbf{V} \setminus \{Z\}$. To see $\mathrm{Pr}''_{1x}(\mathbf{Y}) = \mathrm{Pr}'_{1x}(\mathbf{Y})$ and $\mathrm{Pr}''_{2x}(\mathbf{Y}) = \mathrm{Pr}'_{2x}(\mathbf{Y})$, note that summing-out $R$ from $\mathrm{Pr}''_{1x}(\mathbf{V}, R)$ and $\mathrm{Pr}''_{2x}(\mathbf{V}, R)$ yields $\mathrm{Pr}'_{1x}(\mathbf{V})$ and $\mathrm{Pr}'_{2x}(\mathbf{V})$.

**Third Step:** we construct $\langle G, \mathcal{F}'''_1 \rangle$, $\langle G, \mathcal{F}'''_2 \rangle$ from $\langle G'', \mathcal{F}''_1 \rangle$, $\langle G'', \mathcal{F}''_2 \rangle$ by merging the auxiliary root variable $R$ with an observed parent $T$ of $Z$. We merge $R$ and $T$ into a new node $T'$ and substitute it for $T$ in $G$, i.e., $T'$ has the same parents and children as $T$ in $G$. Specifically, $T'$ is constructed as the Cartesian product of $R$ and $T$: each state of $T'$ can be represented as $(r, t)$ where $r$ is a state of $R$ and $t$ is a state of $T$. We then assign the CPT $f'''_{T'}(T'|\mathbf{P}_T)$ in $\mathcal{F}'''_1$ as follows. For each parent instantiation $\mathbf{p}_{T'}$ and each state $(r, t)$ of $T'$, $f'''_{T'}((r, t)|\mathbf{p}_{T'}) = f''_R(r) f''_T(t|\mathbf{p}_{T'})$. For each child $C$ of $T'$ that has parents $\mathbf{P}_C$ (excluding $T'$), the CPT for $C$ in $\mathcal{F}'''_1$ is assigned as $f'''_C(c|\mathbf{p}_C, (r, t)) = f''_C(c|\mathbf{p}_C, t)$. Similarly, we assign the CPTs $g'''_{T'}$ and $g'''_C$ in $\mathcal{F}'''_2$. The distributions over observed variables are preserved since there is an one-to-one correspondence between the instantiations over $\mathbf{V} \cup \{R\}$ in $\langle G'', \mathcal{F}''_1 \rangle$ and the instantiations over $\mathbf{V}$ in $\langle G, \mathcal{F}'''_1 \rangle$. We next consider the causal effect. Suppose $T$ is neither a treatment nor an outcome variable, then the merging does not affect the causal effect by the one-to-one correspondence between instantiations and $\mathrm{Pr}'''_{1x}(\mathbf{Y}) = \mathrm{Pr}''_{1x}(\mathbf{Y}) \neq \mathrm{Pr}''_{2x}(\mathbf{Y}) = \mathrm{Pr}'''_{2x}(\mathbf{Y})$. Suppose $T$ is an outcome variable, since $\mathrm{Pr}''_{1x}(\mathbf{Y}) \neq \mathrm{Pr}''_{2x}(\mathbf{Y})$, there exists an instantiation $(\mathbf{y}', r, t)$ where $\mathbf{Y}' = \mathbf{Y} \setminus \{T\}$ such that $\mathrm{Pr}''_{1x}(\mathbf{y}', r, t) \neq \mathrm{Pr}''_{2x}(\mathbf{y}', r, t)$.

This implies $\Pr_{1x}'''(\mathbf{y}', (r, t)) \neq \Pr_{2x}'''(\mathbf{y}', (r, t))$ for the particular instantiation $\mathbf{y}'$ and the state $(r, t)$ of $T'$. Suppose $T$ is the treatment variable $X$, since $\Pr_{1x}''(\mathbf{Y}) \neq \Pr_{2x}''(\mathbf{Y})$, there exists an instantiation $(\mathbf{y}, r)$ such that $\Pr_{1x}''(\mathbf{y}, r) \neq \Pr_{2x}''(\mathbf{y}, r)$. Moreover, $\Pr_{1x}''(r) = \Pr_1''(r) = \Pr_2''(r) = \Pr_{2x}''(r) > 0$ (otherwise, $\Pr_{1x}''(\mathbf{y}, r) = \Pr_{2x}''(\mathbf{y}, r) = 0$). This implies $\Pr_{1x}''(\mathbf{y}|r) \neq \Pr_{2x}''(\mathbf{y}|r)$. Since $R$ is a root in the mutilated CBN, $\Pr_{1(xr)}''(\mathbf{y}) = \Pr_{1x}''(\mathbf{y}|r) \neq \Pr_{2x}''(\mathbf{y}|r) = \Pr_{2(xr)}''(\mathbf{y})$. We now consider the treatment $do(T' = (r, x))$ on $G$ instead of the treatment $do(R = r, X = x)$ on $G''$. We have $\Pr_{1((r,x))}'''(\mathbf{y}) = \Pr_{1(r,x)}''(\mathbf{y}) \neq \Pr_{2(r,x)}''(\mathbf{y}) = \Pr_{2((r,x))}'''(\mathbf{y})$ for the particular state $(r, x)$ of $T'$. Moreover, $\Pr_1'''((r, x)) = \Pr_1''(r, x) = \Pr_1''(r)\Pr_1''(x) > 0$ by the positivity assumption of $\Pr_1''(x)$. Thus, the positivity still holds for $\Pr_1'''$ and similarly for $\Pr_2'''$. $\qquad\square$

*Proof of Theorem 18.* Let $\mathbf{H} = \mathbf{V}' \setminus \mathbf{V}$ be the set of hidden functional variables that are functionally determined by $\mathbf{V}$. By Theorem 13, F-identifiable wrt $\langle G, \mathbf{V}, \mathcal{C}_{\mathbf{V}}, \mathbf{W} \rangle$ iff F-identifiable wrt $\langle G', \mathbf{V}, \mathcal{C}_{\mathbf{V}}, \mathbf{W} \setminus \mathbf{H} \rangle$ where $G'$ is the result of functionally eliminating $\mathbf{H}$ from $G$. By construction, every variable in $\mathbf{H}$ has parents in $\mathbf{V}'$. Hence, by Theorem 15, F-identifiable wrt $\langle G', \mathbf{V}, \mathcal{C}_{\mathbf{V}}, \mathbf{W} \setminus \mathbf{H} \rangle$ iff F-identifiable wrt $\langle G, \mathbf{V}', \mathcal{C}_{\mathbf{V}}, \mathbf{W} \rangle$. If we consider each functional variable $W \in \mathbf{W}$, it is either in $\mathbf{V}'$ or having some parent that is not in $\mathbf{V}'$ (otherwise, $W$ would have been added to $\mathbf{V}'$). By Lemma 29, F-identifiable wrt $\langle G, \mathbf{V}', \mathcal{C}_{\mathbf{V}}, \mathbf{W} \rangle$ iff identifiable wrt $\langle G, \mathbf{V}', \mathcal{C}_{\mathbf{V}} \rangle$. $\qquad\square$

