# OpenReview forum: "Identifying Causal Effects Under Functional Dependencies"
_NeurIPS.cc/2024/Conference — NeurIPS 2024 spotlight_

### Official Review · Reviewer_GMHe · 2024-06-12

**Soundness:** 4
**Presentation:** 3
**Contribution:** 3
**Rating:** 8
**Confidence:** 4

**Summary:**

The paper studies the identifiability of causal effects in the presence of functional dependencies. Section 4 starts with an engaging discussion on the interaction between positivity constraints and functional dependencies, during which it defines the concept of F-identifiability. Then in Section 4 its introduces new concepts such as functional elimination, functional projection, and, which are pivotal for much of the subsequent discussion. Additionally, in this section, it presents several results that reduce D-separation to d-separation. In Section 5, the paper presents various results that reduce F-identifiability  to F-identifiability from a simpler graph (Theorems 13 and 15) or to identifiability (Theorems 14, 16, 17 and 18).

**Strengths:**

I think that the paper is addressing an intriguing topic, extending identifiability results to encompass functional dependencies, which could prove valuable in numerous applications. The results are eloquently presented, with examples included in the main paper and complete proofs provided in the Appendix, making the paper comprehensible for readers acquainted with the subject matter.

**Weaknesses:**

* Explaining how the theoretical results of this paper can be practically applied to real-world datasets can make the theoretical results in this paper more relevant (I think the results are relevant and important for real world applications).  See Questions below.

* The paper presents numerous theorems that share a similar objectives. I wonder if it's possible to consolidate them into two general, elegant rules, one one for reducing F-identifiability to F-identifiability from simpler graphs and one for reducing F-identifiability to identifiability. However, I don't believe this issue alone warrants rejection. Each theorem, although not unified under a single rule, provides unique insights in different contexts, all of which have both theoretical and practical value.

**Questions:**

Minor:
* Can you add a new section for example titled “Motivating Example” to present a hypothetical case study where knowledge of functional variables leads to successful identification, whereas classical identification methods would have suggested unidentifiability.
* I think the sentence at line 216 can be made clearer by avoiding a double "if", for example: If every positivity constraint that mentions W does not mention HW ...
* To my knowledge, a Bayesian network is not necessarily a causal graph. Therefore, in certain places such as footnote 3, it might be more accurate to replace "Bayesian network" with "causal Bayesian network" to emphasize that the graph should be causal.

**Limitations:**

The authors have addressed the limitations, but I believe they could put additional effort into emphasizing them. For example, they could highlight which theorems can provide an identifying formula and which cannot. Furthermore, clearly (in conclusion or in a dedicated paragraph) discussing future work in light of these limitations would be beneficial.

---

> ### Author Rebuttal · Authors · 2024-08-06
>
> > The paper presents numerous theorems that share a similar objectives. I wonder if it's possible to consolidate them into two general, elegant rules, one one for reducing F-identifiability to F-identifiability from simpler graphs and one for reducing F-identifiability to identifiability. However, I don't believe this issue alone warrants rejection. Each theorem, although not unified under a single rule, provides unique insights in different contexts, all of which have both theoretical and practical value.
>
> Thanks for the feedback! We will see if there is a succinct way to summarize these rules.
>
> > Explaining how the theoretical results of this paper can be practically applied to real-world datasets can make the theoretical results in this paper more relevant (I think the results are relevant and important for real world applications). See Questions below.
> > Can you add a new section for example titled “Motivating Example” to present a hypothetical case study where knowledge of functional variables leads to successful identification, whereas classical identification methods would have suggested unidentifiability.
>
> Thanks for the suggestion. We will add the following hypothetical example to the introduction. We want to study how the enforcement of speed limits affects the accident rate. According to our knowledge, the legal driving age (A) is functionally determined by the country (C), and both legal driving age and country are causes of driving speed (X). Moreover, the driving speed and legal driving age are causes of accidents (Y). We can construct a causal graph with edges $C \rightarrow A, C \rightarrow X, A \rightarrow X, A \rightarrow Y, X \rightarrow Y$. Suppose we observe variables $\{C, X, Y\}$, the classical identification method will suggest that the causal effect of X on Y is unidentifiable. However, it is in fact F-identifiable given that A is functionally determined by C.
>
> > I think the sentence at line 216 can be made clearer by avoiding a double "if", for example: If every positivity constraint that mentions W does not mention HW …
>
> OK, will fix it. Thanks!
>
> > To my knowledge, a Bayesian network is not necessarily a causal graph. Therefore, in certain places such as footnote 3, it might be more accurate to replace "Bayesian network" with "causal Bayesian network" to emphasize that the graph should be causal.
>
> Yes, We are aware that Bayesian networks may not be causal. We tried to save space by omitting (causal) as shown on Line 72, but we will add it back to avoid confusion.
>
> > The authors have addressed the limitations, but I believe they could put additional effort into emphasizing them. For example, they could highlight which theorems can provide an identifying formula and which cannot. Furthermore, clearly (in conclusion or in a dedicated paragraph) discussing future work in light of these limitations would be beneficial.
>
> Thank you for the suggestion. Will do.

---

> > ### Comment · Reviewer_GMHe · 2024-08-10
> >
> > Thank you for your response on my minor remarks. After considering the other reviews, I continue to find this paper very interesting and will maintain my original score.

---

### Official Review · Reviewer_3A71 · 2024-07-09

**Soundness:** 4
**Presentation:** 3
**Contribution:** 3
**Rating:** 7
**Confidence:** 3

**Summary:**

The paper addresses a novel problem in causal effect identifiability, introducing the concept of functional dependency among variables. It proposes a new elimination approach for removing redundant variables from the graph while preserving the identifiability of the target quantity. The main contribution includes proposing graph conditions that reduce the problem to classic ID settings.

**Strengths:**

1- The paper addresses a novel problem in causal effect identifiability.

2- The paper is well-written, and the authors formulate the problem clearly.

3-  An approach called functional elimination has been proposed to remove redundant variables while preserving identifiability.

4- By leveraging functional elimination and certain graph conditions, the theorems demonstrate that F-Identifiability can be reduced to the classic ID problem.

**Weaknesses:**

The proposed conditions are not complete; the paper does not provide a complete condition to determine whether a causal effect is id or not.

**Questions:**

1-  I didn’t understand when your theorems fail to recognize whether a causal effect in a graph is id. Can you provide some examples where the conditions of Theorem 15 are not satisfied, but the causal effect is identifiable (or not)?


2- Could you elaborate on corollary 14 and the positivity constraint? It’s worth rechecking the exact conditions of projection operation and stating the corollary in precise form.


3- What about the generalization of results to other variants of the causal effect identification problem, such as c-ID, g-ID, s-ID, and so on? If there is additional space, it would be beneficial to include a brief review of these works.

**Limitations:**

Yes.

---

> ### Author Rebuttal · Authors · 2024-08-06
>
> > I didn’t understand when your theorems fail to recognize whether a causal effect in a graph is id. Can you provide some examples where the conditions of Theorem 15 are not satisfied, but the causal effect is identifiable (or not)?
>
> We suspect that Theorem 15 will hold under weaker positivity constraints, but the current condition is the most succinct one we can obtain so far. The difficulty lies in how to formulate the positivity constraints $C’_V$ for the graph resulting from eliminating functional variable $Z$ if the original positivity constraints $C_V$ contain $Z$. For example, we cannot simply set $C’_V$ to be the positivity constraints in $C_V$ that do not mention $Z$. To see why, consider a causal graph with observed variables $A,Z,X,Y$ and edges $A \rightarrow Z, Z \rightarrow X, Z \rightarrow Y, X \rightarrow Y$. Assume that variable $Z$ is functional and the positivity constraint is $Pr(X|Z) > 0$. The causal effect of $X$ on $Y$ is F-identifiable. However, it is no longer F-identifiable in the graph resulting from eliminating $Z$ by Proposition 5 since $C’_V = \emptyset$ if we only keep the positivity constraints that do not mention $Z$.
>
> > Could you elaborate on corollary 14 and the positivity constraint? It’s worth rechecking the exact conditions of projection operation and stating the corollary in precise form.
>
> In [Tian & Pearl, 2002], the projection operation is stated under the strict positivity $Pr(V) > 0$. We are saying that if projection also operates under a weaker positivity assumption, then we can relax the positivity constraint accordingly in Corollary 14. We will clarify this further.
>
> > What about the generalization of results to other variants of the causal effect identification problem, such as c-ID, g-ID, s-ID, and so on? If there is additional space, it would be beneficial to include a brief review of these works.
>
> Thank you for the suggestion! We will consider it.

---

> > ### Comment · Reviewer_3A71 · 2024-08-09
> >
> > Thank you for your response. After considering the feedback from other reviewers and your replies, I have revised my evaluation to 7.

---

### Official Review · Reviewer_bXg6 · 2024-07-10

**Soundness:** 4
**Presentation:** 3
**Contribution:** 4
**Rating:** 8
**Confidence:** 4

**Summary:**

Existing causal effect identification algorithms such as the ID algorithm, require strict positivity constraints on the observed distribution that can get violated in cases where some variables (observed or hidden) are deterministic functions of their parents. This paper takes a step towards finding conditions for when causal effects can be identified when such 'functional variables' are present (with quantitative knowledge of functional dependencies being not required). At a high-level, the approach is to eliminate functional variables followed by latent projection that enables using existing ID algorithms on the resulting causal Bayesian nets. Sound conditions for identifiability are proposed that depend on whether the functional variables are hidden or observed.

**Strengths:**

It's quite clear that functional variables are ubiquitous in the world and hence causal modeling invariably encounters such variables. The dependence of do-calculus and ID algorithm on strict positivity is a hindrance to identify causal effects in such models that include functional variables. This is a nice first step in causal effect identification in the presence of functional variables. I like the view of identification algorithms that take as input positivity constraints which is further extended to taking the functional variables as input. The paper is clearly written overall with some minor issues mentioned in the weakness section. While, the relationship between positivity constraints and functional dependencies was not fully explored, the necessary condition for identifiability w.r.t. the positivity constraints is a nice addition. The proofs of the main theorems are correct but I have not completely checked the proofs of the intermediate lemmas.

This work also opens up multiple avenues for future research into developing complete identification algorithms and perhaps inspires work into ID-style algorithms that take in the weakest possible positivity constraints. This seems an active thread of investigation already, see for example, "On Positivity Condition for Causal Inference" accepted to ICML 2024.

**Weaknesses:**

I believe the current version is dense with results with the page restriction limiting a better style of presentation. There are multiple corollaries that deserve to be highlighted separately that appear in the middle of the text. I would also prefer adding proof sketches of the main theorems and cutting out results that don't directly impact the main message of the paper, for example, perhaps limiting the section on positivity contraints since it appears as an interlude in the current version.

**Questions:**

A few questions and typos:
1) Line 169 - the constraint is sufficient for identification under the assumption that there exists an instantiation of Z such that Pr(z)>0. This is also later used in the proof of Proposition 5. Why is this assumption justified?
2) The footnote remark on Page 8 is ambiguous. What in Ref 25,26 point to requiring the positivity condition for projection?
3) Proof of Proposition 5 - typo in definition of f^1(z|x,p_Z)

**Limitations:**

Limitations of the technical content aren't explicitly discussed. Potential negative societal impact is not applicable.

---

> ### Author Rebuttal · Authors · 2024-08-06
>
> > I believe the current version is dense with results with the page restriction limiting a better style of presentation. There are multiple corollaries that deserve to be highlighted separately that appear in the middle of the text. I would also prefer adding proof sketches of the main theorems and cutting out results that don't directly impact the main message of the paper, for example, perhaps limiting the section on positivity contraints since it appears as an interlude in the current version.
>
> Thank you for your suggestions! We will consider them.
>
> > Line 169 - the constraint is sufficient for identification under the assumption that there exists an instantiation of Z such that Pr(z)>0. This is also later used in the proof of Proposition 5. Why is this assumption justified?
>
> This is a relaxed version of positivity from the ID algorithm [Shpitser & Pearl, 2006], which requires $Pr(X|P) > 0$, where P are the observed parents of X. In this example, we only need the assumption $Pr(X|Z) > 0$ since it is sufficient to make the identifying formula well-defined. This is because $Pr(y|x, z) Pr(z)$ in the formula is equal to zero when $Pr(z)=0$, and is computable when $Pr(z) > 0$ (the conditional probability $Pr(y|x,z)$ is well-defined if $Pr(x|z) > 0$).
>
> > The footnote remark on Page 8 is ambiguous. What in Ref 25,26 point to requiring the positivity condition for projection?
>
> Ref 26 is more relevant in this case. Ref 25 mainly focuses on d-separations. We will fix this.
> Thank you for pointing it out!
>
> > Proof of Proposition 5 - typo in definition of f^1(z|x,p_Z)
>
> We will fix it. Thanks!

---

> > ### Comment · Reviewer_bXg6 · 2024-08-13
> > **Response to rebuttal**
> >
> > Thanks for the response. I am happy to maintain my original score.

---

### Official Review · Reviewer_uRPz · 2024-07-12

**Soundness:** 4
**Presentation:** 3
**Contribution:** 3
**Rating:** 7
**Confidence:** 3

**Summary:**

This paper investigates the identification of causal effects in the presence of functional dependencies, where some variables are determined by their parents. The study demonstrates that unidentifiable causal effects can become identifiable and that certain functional variables can be excluded from observation without affecting identifiability. The authors introduce a new elimination procedure to remove functional variables while preserving key properties of the causal graph, and show how existing algorithms can be used to test and obtain identifying formulas. This approach can significantly reduce the number of variables needed in observational data.

**Strengths:**

1. The paper is written well and easy to understand
2. Theoretical aspects are explained with the help of examples which makes it easy to follow along.

**Weaknesses:**

1. What are the implications when the treatment/target variable is a functional variable?
2. Assumptions need to be formally written. Currently, many assumptions are not very clear. For example, are functional variables need to be observed or they can also be unobserved?
3. Justification for NeurIPS checklist items is not provided where necessary.

**Questions:**

See weaknesses section

**Limitations:**

Limitations are not discussed

---

> ### Author Rebuttal · Authors · 2024-08-06
>
> > What are the implications when the treatment/target variable is a functional variable?
>
> If each treatment and target variable has some hidden parent and these are the only functional variables (after perhaps eliminating other functional variables by Theorems 13 & 15), we can reduce F-identifiability to classical identifiability by Theorem 17 (we will include a note to this effect). Otherwise, our current results do not cover this case which is a subject for future work.
>
> > Assumptions need to be formally written. Currently, many assumptions are not very clear. For example, are functional variables need to be observed or they can also be unobserved?
>
> We allow both observed and unobserved (hidden) functional variables in the causal graph. This is evident by Theorem 13 & Corollary 14 which eliminate hidden functional variables, and Theorem 15 & Corollary 16 which eliminate observed functional variables. We would really appreciate it if you could point us to any unclear assumptions so that we can clarify them further.
>
> > Justification for NeurIPS checklist items is not provided where necessary.
>
> We did not see ambiguity in our answers to the checklist items, but we are happy to add justifications to these items. Please let us know in case any of them need particular attention and we will address them properly.

---

> > ### Comment · Reviewer_uRPz · 2024-08-12
> > **Thank you for your response.**
> >
> > I thank the authors for their response. I've read their response and I will increase my score. Regrading checklist items, I suggest writing some justifications for [YES] or [NA] instead of leaving them to TODO.

---

### Author Rebuttal · Authors · 2024-08-06

We thank the reviewers for their thoughtful comments and suggestions. Please see individual responses below.

---

### Decision · Program_Chairs · 2024-09-25

**Decision:**

Accept (spotlight)

**Comment:**

This paper explores the identification of causal effects in the presence of functional dependencies, where some variables are determined by their parents, introducing an elimination procedure to handle functional variables. The study demonstrates that unidentifiable causal effects can become identifiable and proposes conditions under which existing algorithms can be adapted for this purpose.

Reviewers unanimously recognized its novelty and theoretical contributions while some of the reviewers suggested clarifications and additional examples to better illustrate the practical implications. I recommend it for acceptance and encourage the authors to address the reviewers minor comments such as emphasizing the limitations of this work and discussing potential future research directions in light of these limitations. Additionally, please discuss how this work relates to a recent paper published at ICML, "On Positivity Condition for Causal Inference" by Hwang et al. (2024).